# Intra-Cluster Mixup: An Effective Data Augmentation Technique for Complementary-Label Learning

**Tan-Ha Mai**  *d10922024@csie.ntu.edu.tw*
*Department of Computer Science and Information Engineering,*
*National Taiwan University*

**Hsuan-Tien Lin**\*  *htlin@csie.ntu.edu.tw*
*Department of Computer Science and Information Engineering,*
*Artificial Intelligence Center of Research Excellence,*
*National Taiwan University*

**Reviewed on OpenReview:** *https://openreview.net/forum?id=h9PbmfznWj*

## Abstract

In this paper, we investigate the challenges of *complementary-label learning (CLL)*, a specialized form of *weakly-supervised learning (WSL)* where models are trained with labels indicating classes to which instances do not belong, rather than standard ordinary labels. This alternative supervision is appealing because collecting complementary labels is generally cheaper and less labor-intensive. Although most existing research in CLL emphasizes the development of novel loss functions, the potential of data augmentation in this domain remains largely underexplored. In this work, we uncover that the widely-used Mixup data augmentation technique is ineffective when directly applied to CLL. Through in-depth analysis, we identify that the complementary-label noise generated by Mixup negatively impacts the performance of CLL models. We then propose an improved technique called *Intra-Cluster Mixup (ICM)*, which only synthesizes augmented data from nearby examples, to mitigate the noise effect. ICM carries the benefits of encouraging complementary label sharing of nearby examples, and leads to substantial performance improvements across synthetic and real-world labeled datasets. In particular, our wide spectrum of experimental results on both balanced and imbalanced CLL settings justifies the potential of ICM in allying with state-of-the-art CLL algorithms, achieving significant accuracy increases of 30% and 10% on MNIST and CIFAR datasets, respectively.

## 1 Introduction

Obtaining high-quality labels is often expensive, time-consuming, and sometimes impossible in real-world applications. To address this challenge, *weakly-supervised learning (WSL)* has been extensively studied in recent years. WSL aims to train a proper classifier with inaccurate, incomplete, or inexact supervision (Zhou, 2017). Contemporary WSL studies have significantly expanded our understanding of machine learning capabilities, encompassing areas such as learning from complementary labels (Ishida et al., 2017), learning from multiple complementary labels (Feng et al., 2020), learning from partial labels (Jin & Ghahramani, 2002), or a mixture of ordinary and complementary labels (Ishida et al., 2017).

This paper focuses on *complementary-label learning (CLL)* (Ishida et al., 2017), a WSL problem where a complementary label designates a class to which a specific instance *does not belong*. The CLL problem assumes that the learner only has access to complementary labels during training while still expecting the learner to predict the ordinary labels correctly during testing. Complementary labels serve as a viable alternative when it is difficult or costly to acquire ordinary labels (Ishida et al., 2017). For instance, collecting

---

\*Corresponding author.

ordinary labels on numerous classes not only demands annotators with excellent knowledge for selecting the correct labels but also requires more time for accurate labeling. CLL models can extend the horizon of machine learning and make multi-class classification potentially more realistic when ordinary labels cannot be easily obtained (Ishida et al., 2017).

Existing CLL studies have primarily focused on designing loss functions that are converted from well-known ordinary classification losses (Ishida et al., 2017; 2018), often under the assumption that complementary labels are uniformly generated (Cao et al., 2022; Feng et al., 2020). Building on this line of work, (Wang et al., 2024) recently proposed a novel data generation assumption inspired by positive-unlabeled learning, offering a fresh perspective on how complementary labels may be distributed. Collectively, these loss-function-based approaches address the CLL problem from an *algorithmic* perspective and significantly contribute to our understanding of the design space for CLL models. Despite this focus, the potential of data augmentation in CLL remains largely unexplored. This notable gap in the literature motivates our study on developing and assessing data augmentation techniques to improve the efficacy of CLL models.

Data augmentation techniques are known to be powerful *"add-ons"* to machine learning models for enhancing their performance by improving generalization, robustness to noise, and invariance to transformations (Mikołajczyk & Grochowski, 2018; Rebuffi et al., 2021). Across a range of classification scenarios (Chou et al., 2020a), successful data augmentation techniques exhibit seamless integration with algorithmic approaches to boost their performance. Some data augmentation techniques create pseudo examples that are variations of the original examples, without re-labeling them (Shorten & Khoshgoftaar, 2019; Jiang et al., 2021). Other techniques construct synthetic examples with modified labels (Lin & Lin, 2023). Motivated by recent studies on multiple complementary-label learning (Cao et al., 2022; Feng et al., 2020), we conjecture that utilizing multiple complementary labels through label sharing has the potential to improve existing CLL models, and thus resort to label-modification techniques. Among such techniques, we choose Mixup (Zhang et al., 2018) because of its natural potential in encouraging complementary-label sharing. While Mixup is well-known for its simplicity and effectiveness (Li & Jia, 2025; Navarro & Segarra, 2023; Xie et al., 2023), the application of Mixup for CLL remains unexplored prior to our work.

With a wide spectrum of experiments across balanced and imbalanced CLL settings, we confirm that applying Mixup on the complementary labels has the potential to improve various state-of-the-art CLL models by encouraging label sharing, which helps the machine identify the ordinary label more efficiently (Lin & Lin, 2023; Chou et al., 2020b; Yu et al., 2018). But the potential comes with a serious side effect. In particular, sometimes the complementary labels on which Mixup manipulates contains an *ordinary* label of one of the examples, which introduces noise to the label sharing process. The noise significantly deteriorates the performance of the CLL model because of overfitting. The side effect suggests that original Mixup does not work off-the-shelf for CLL.

To mitigate the side effect, we design a novel data augmentation technique called *Intra-Cluster Mixup (ICM)*. ICM clusters the examples before applying Mixup *within each cluster*. The clustering design reduces the noise introduced by Mixup while keeping its potential benefits. Our empirical experiments demonstrate that ICM consistently enhances the CLL performance across a variety of state-of-the-art CLL models and a broad spectrum of settings, ranging from balanced to imbalanced classification. Furthermore, we expand our empirical comparison from 4 common benchmark datasets in existing studies to 7, including both synthetic and real-world labeled datasets. Our efforts significantly broaden the scope of benchmarks in the field. Our unique contributions can be summarized as follows:

- To the best of our knowledge, we are *the first* to introduce a novel data augmentation technique specifically designed for CLL contexts. We identify two critical insights: (i) the original Mixup fails in CLL settings due to noise introduced during the label sharing process, and (ii) mixing samples within the same class proves to be a more effective strategy.

- We propose ICM, a tailored data augmentation technique that addresses the unique challenges of CLL and consistently enhances the performance across various CLL models.

- We conduct extensive benchmarking on large CLL datasets, covering a range from *synthetic* to *real-world* labeled datasets. Our studies span a diverse spectrum of settings, from *balanced* to *imbalanced* CLL, justifying the effectiveness of our framework.

## 2 Problem Setup

### 2.1 Complementary-Label Learning

In CLL, we are given a dataset $\bar{D} = \{(\mathbf{x}_i, \bar{y}_i)\}_{i=1}^{N}$, where each instance $\mathbf{x}_i \in \mathbb{R}^d$ represents an input image, and $\bar{y}_i \in \mathbb{R}^K$ is a complementary label. The complementary label $\bar{y}_i$ indicates a class that the image $\mathbf{x}_i$ does not belong. The dataset consists of $N$ samples, and the goal of CLL is to use this complementary label information to train a classifier. In this context, the complementary label satisfies the condition $\bar{y}_i \in [K] \backslash \{y_i\}$, where $y_i$ is the ordinary label of $\mathbf{x}_i$, $K$ denotes the total number of classes in the dataset, with $K > 2$. The set $[K] = \{1, 2, \ldots, K\}$ represents the set of all possible class labels. This implies that $\bar{y}_i$ is one of the incorrect classes for the instance $\mathbf{x}_i$. The training set $\bar{D}$ is denoted as $\bar{D} = X \times \bar{Y}$, where $X$ contains the input images and $\bar{Y}$ contains the corresponding complementary labels. In contrast to traditional multi-class classification, where the ordinary label $y_i$ is used to train a classifier, the CLL setup trains the model using the complementary label $\bar{y}_i$. However, the objective in CLL remains the same: to train a classifier $g$ that accurately predicts the ordinary label $y_i$ for unseen instances. Generally, the classifier $g$ is realized through a decision function $g \colon \mathbb{R}^d \to \mathbb{R}^K$, with the classification determined by taking the argmax on $h$. For example, $g(\mathbf{x}) = \mathrm{argmax}_{k \in [K]} h(\mathbf{x})_k$, where $h(\mathbf{x})_k$ represents the score or confidence that the instance $\mathbf{x}$ belongs to class $k$. The classifier selects the class with the highest score.

### 2.2 Recent Approaches of Complementary-Label Learning

Recent approaches to CLL share a common characteristic: they apply various surrogate loss functions to the standard classifier. For instance, (Ishida et al., 2017; 2019) developed an *unbiased risk estimator (URE)* for arbitrary losses on the standard classifier when employing a uniform transition matrix. When the risk is defined as the classification error, the URE serves as a surrogate metric for performance evaluation. However, UREs are prone to severe negative empirical risks during training, which is indicative of overfitting. To mitigate such overfitting in algorithm design, (Chou et al., 2020b) proposed the *surrogate complementary loss (SCL)*, which is based on minimizing the likelihood associated with complementary label. They justified this approach by showing that SCL constitutes an upper bound to a constant multiple of the standard classification error when the transition matrix is uniform. Another study by (Yu et al., 2018) examined scenarios where the complementary label are not uniformly generated. (Yu et al., 2018) introduced a framework called *forward correction (FWD)* that adapts techniques from noisy label learning (Patrini et al., 2017) to adjust the softmax cross-entropy loss. This is achieved by adding a transition layer to the output of the model: $\bar{g}(\mathbf{x}) = T^T g(\mathbf{x})$, and then utilizing the cross-entropy loss between $\bar{g}(\mathbf{x})$ and the complementary labels $\bar{y}$. Other research efforts have explored advanced applications beyond single complementary label, including learning from multiple complementary labels (Cao et al., 2022; Feng et al., 2020), and integrating learning from both ordinary and complementary labels (Katsura & Uchida, 2020).

## 3 Proposed Method

In this section, we propose ICM, a novel data-augmentation technique for CLL. First, we evaluate the performance of the standard Mixup method under various experimental conditions to identify the factors that undermine its effectiveness. Next, we develop enhanced augmentation algorithms that explicitly address these limitations. Finally, we derive and introduce a surrogate complementary-label loss function that seamlessly integrates ICM into the training process.

### 3.1 Why Mixup does not work?

Applying Mixup naively in the CLL setting results in substantial *complementary-label noise*. This noise arises when the ordinary label appears in the synthetic data generated via original Mixup, thereby violating

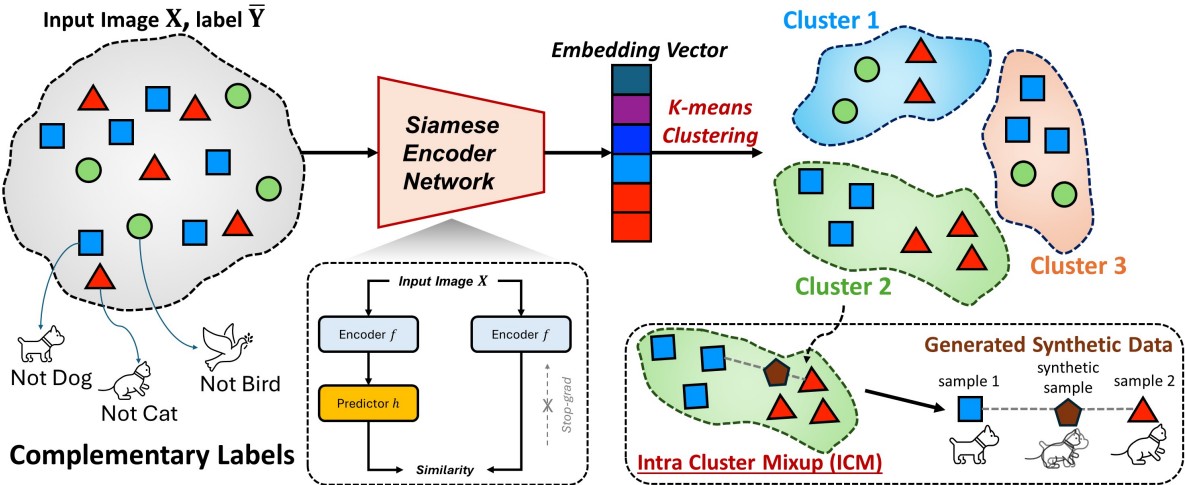

Figure 1: Illustration of the *Intra-Cluster Mixup (ICM)* framework. *Top*: Embedding features are extracted using a pretrained *SimSiam* encoder and clustered using $k$-means, aiming to group samples with similar ordinary labels. *Bottom right*: Within each cluster, ICM generates synthetic samples by interpolating features and labels, which are then used to train the classifier.

the core assumption of CLL: $\bar{y}_i \in [K] \setminus y_i$. To empirically verify this claim, we conduct an ablation study measuring the noise ratio introduced by Mixup in a controlled CLL setting. Although CLL typically operates under the assumption that ordinary labels are unavailable or costly to obtain, we adopt a *proof-of-concept* setup where ordinary labels are accessible solely for quantifying the noise. Using the SCL-NL loss (Chou et al., 2020b) and a ResNet18 backbone (He et al., 2016) trained on CIFAR10, we observe that Mixup introduce a noise level of 15.81% (as indicated by the green triangle in Figure 2a). Notably, when training is performed under noise-free conditions, model accuracy improved by 7%, indicating high sensitivity to label noise (highlighted by the orange circle in Figure 2a). These results highlight that label noise in CLL substantially degrades performance. From these observations, we introduce a mathematical framework for analyzing complementary classification error under Mixup augmentation.

**Definition 1** (Complementary classification error). Let $\{(\mathbf{x}_i, \bar{y}_i)\}_{i=1}^N$ be the training examples, where $\mathbf{x}_i \in \mathbb{R}^d$ is the input and $\bar{y}_i \in \{1, \ldots, K\}$ is the complementary hard label of the $i$-th example. Let $g : \mathbb{R}^d \to \{1, \ldots, K\}$ be a classifier and let $\ell(\cdot, \cdot)$ denote a loss function. For any input $\mathbf{x}$, define the per-class loss vector $\ell\big(g(\mathbf{x})\big) = \big[\ell\big(1, g(\mathbf{x})\big), \ldots, \ell\big(K, g(\mathbf{x})\big)\big]$. Given two training samples $\mathbf{x}_i$ and $\mathbf{x}_j$ from the same cluster ($j$ is an index randomly sampled from the same cluster as $i$), we construct a mixed input $\tilde{\mathbf{x}}_{i,j} = \lambda \mathbf{x}_i + (1-\lambda)\mathbf{x}_j$, where $\lambda \sim \text{Beta}(\alpha, \alpha)$ and $\tilde{y}_{i,j}$ denotes the corresponding soft label generated via Mixup. The complementary classification error of $g$ under loss $\ell$ is defined as

$$\mathcal{R}_{hl}(g; \ell) = \frac{1}{N} \sum_{i=1}^N \ell\big(\bar{y}_i, g(\mathbf{x}_i)\big) = \mathbb{E}_{(\mathbf{x}, \bar{y}) \sim \bar{D}}\big[\, [\![\bar{y} \neq g(\mathbf{x})]\!]\,\big], \quad {}^1 \tag{1}$$

For Mixup-generated pairs $(\tilde{\mathbf{x}}_{i,j}, \tilde{y}_{i,j})$, the complementary classification risk under soft labels is defined as

$$\mathcal{R}_{sl}(g; \ell) = \frac{1}{N} \sum_{i=1}^N \ell\Big(\tilde{y}_{i,j}, g(\tilde{\mathbf{x}}_{i,j})\Big) = \mathbb{E}_{(\mathbf{x}, \bar{y}) \sim \bar{D}} [\![\bar{y}_i \neq g(\tilde{\mathbf{x}}_{i,j})]\!] + \mathbb{E}_{(\mathbf{x}, \bar{y}) \sim \bar{D}} [\![\bar{y}_j \neq g(\tilde{\mathbf{x}}_{i,j})]\!]. \tag{2}$$

**Definition 2** (Error generated by label noise). The error generated by label noise for the classifier $g$ is defined as

$$\varepsilon(g) = \mathbb{E}_{(\mathbf{x}, \bar{y}) \sim \bar{D}} [\![\bar{y} = g(\mathbf{x})]\!], \tag{3}$$

that is, $\varepsilon(g)$ is the probability that $g$ predicts the complementary label itself.

---

[1]Here, $[\![\cdot]\!]$ denotes the indicator function: for any condition $A$, $[\![A]\!] = 1$ if $A$ holds and 0 otherwise.

**Proposition 1** (Complementary error with Mixup). For Mixup-generated pairs $(\tilde{\mathbf{x}}_{i,j}, \tilde{y}_{i,j})$, the complementary classification risk under Mixup is

$$\mathcal{R}'(g;\ell) = \frac{1}{N}\sum_{i=1}^{N}\ell\big(\tilde{y}_{i,j}, g(\tilde{\mathbf{x}}_{i,j})\big), \tag{4}$$

and admits the decomposition

$$\mathcal{R}'(g;\ell) = \lambda\mathbb{E}_{(\mathbf{x},\bar{y})\sim\bar{D}}\left[\!\left[\bar{y}_i \neq g(\tilde{\mathbf{x}}_{i,j})\right]\!\right] + (1-\lambda)\mathbb{E}_{(\mathbf{x},\bar{y})\sim\bar{D}}\left[\!\left[\bar{y}_j \neq g(\tilde{\mathbf{x}}_{i,j})\right]\!\right] + \lambda\varepsilon_i + (1-\lambda)\varepsilon_j. \tag{5}$$

where $\varepsilon_i$ and $\varepsilon_j$ are the local noise errors defined in equation 3, and satisfy $\varepsilon(g) = \frac{1}{N}\sum_{i=1}^{N}\varepsilon_i$. Thus, the Mixup risk $\mathcal{R}'(g;\ell)$ consists of two classification-error terms weighted by $\lambda$ and $(1-\lambda)$, plus the corresponding contributions from the local label-noise errors of the samples participating in the Mixup pair.

*Proof.* Refer to Appendix A for the proof. □

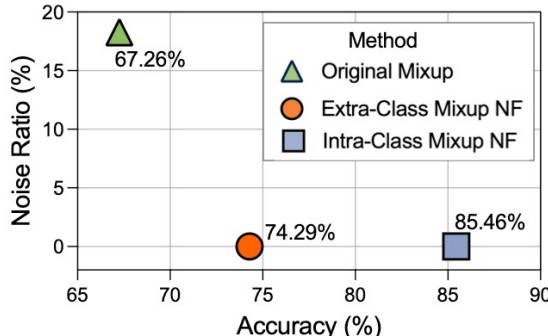

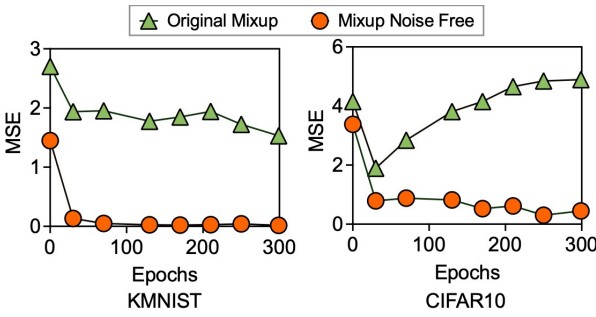

(a) Relationship between noise ratio and model performance on CIFAR-10 with the SCL-NL loss and ResNet18 when applying original Mixup, Extra-Class Mixup Noise-Free (NF) and Intra-Class Mixup NF Increasing the noise ratio in original Mixup degrades model performance. Further ablation study reveals that same class Intra-Class Mixup NF can be more beneficial.

(b) Comparison of gradient estimation errors between original Mixup and *Mixup Noise-Free* on MNIST and CIFAR10, using the SCL-NL loss function and ResNet18 architecture. *Mixup Noise-Free* demonstrates lower gradient estimation error than the original Mixup on both datasets, attributed to reduced noise interference, which impacts classifier performance in CLL contexts.

Figure 2: Analysis of the impact of noise and Mixup Noise-Free (NF) on complementary-label learning performance.

The equation 5 emphasizes that minimizing complementary loss under Mixup requires careful control of the noise rate in the generated data. Specifically, it is critical to minimize instances where $\bar{y}_i = y_j \vee \bar{y}_j = y_i$, as such occurrences increase the error term $\varepsilon$ in equation 5.

To further explore this effect, we evaluate gradient estimation error under noisy and noise-free setups. Let $f$ denote the true gradient and $c$ the complementary gradient estimator. The estimation error is defined as $\mathbb{E}_{(\mathbf{x},y,\bar{y})}[(f-c)^2]$. As shown in Figure 2b for MNIST and CIFAR10, the error associated with *Mixup Noise-Free* is consistently lower than that of original Mixup, reinforcing that label noise compromises optimization effectiveness.

Beyond quantifying noise, we investigate whether the original Mixup strategy used in ordinary learning transfers effectively to the CLL setting. In traditional supervised learning, original Mixup interpolates inputs and labels from different classes, which encourages smooth decision boundaries. In contrast, CLL imposes constraints that make such cross-class interpolation problematic. We hypothesize that mixing data within the same class, termed Intra-Class Mixup, preserves the CLL constraint $\bar{y}_i \in [K] \setminus y_i$ and reduces noise. To evaluate this, we synthetically generate intra-class and extra-class samples under a noise-free setup. Results in Figure 2a (highlighted by the *blue square* and *orange circle*) reveal that *Intra-Class Mixup Noise-*

*Free*[2] outperforms *Extra-Class Mixup Noise-Free*[3] by 11%. This significant margin validates our hypothesis: intra-class mixing is more suitable in the CLL context and significantly improves performance.

The experimental results indicate that original Mixup can be effective under noise-free conditions, eliminating noise typically requires access to ordinary labels which contradicting the fundamental assumption of CLL. To address this challenge, we propose a dedicated framework for CLL, referred to as ICM. This framework is specifically designed to reduce synthetic label noise without requiring knowledge of the ordinary label. The following subsection provides a detailed explanation of the ICM framework.

### 3.2 Intra-Cluster Mixup in CLL

As discussed in above subsection, we introduce a specialized design for CLL called ICM, illustrated in Figure 1. Our proposed methodology, ICM, comprises two primary components. First, feature representations are extracted from the training data using a self-supervised learning model based on SimSiam (Chen & He, 2021). These embeddings are then clustered using $k$-means to group samples with similar feature characteristics, as shown in the *top* of Figure 1. This clustering step assigns cluster-based labels to sample and serves as a pre-processing phase. Second, synthetic complementary samples are generated by mixing inputs and labels within the same cluster, as illustrated in the *bottom right* of Figure 1. These augmented samples are then used to train the classifier. The procedure is defined as:

$$\tilde{\mathbf{x}}_{i,j} = \lambda \mathbf{x}_i + (1 - \lambda)\mathbf{x}_j \tag{6}$$

$$\tilde{y}_{i,j} = \lambda \bar{y}_i + (1 - \lambda)\bar{y}_j. \tag{7}$$

Integration of ICM into the training process is detailed in Algorithm 1. After clustering the dataset $\bar{D}$, ICM selects pairs within the same cluster to generate synthetic complementary samples using equation 6 and equation 7. Here, $\lambda$ is sampled from a beta distribution $\beta(\alpha, \alpha)$, and the selected pairs $(\mathbf{x}_i, \bar{y}_i)$ and $(\mathbf{x}_j, \bar{y}_j)$ are drawn uniformly from the training data.

Clustering plays a critical role by reducing label noise during data augmentation. Grouping samples within clusters encourages mixing between instances that are more likely to share the same true label. This increases the likelihood that the complementary label condition $\bar{y}_i \in [K] \setminus y_i$ holds across the cluster, thereby reducing the risk of introducing noise. To further investigate the effect of encoder choice in our method, we conduct an ablation study comparing SimSiam with other self-supervised encoders; detailed results are provided in Appendix D.7.

To evaluate ICM, we conduct an ablation study comparing it with original Mixup. As shown in Figure 3a, ICM significantly reduces the noise ratio. For instance, on MNIST, the ratio drops from 16.24% with Mixup to 0.95% with ICM. The effectiveness of noise reduction correlates with dataset complexity; simpler datasets such as MNIST, KMNIST, and FMNIST exhibit greater improvements than more complex datasets like CIFAR10 and CIFAR20. This reduction in noise ratio is mirrored by substantial performance improvements. As shown in Figure 3b, ICM consistently outperforms the original Mixup across all algorithms, with gains ranging from 13% to 20%. These results validate the benefit of incorporating clustering into original Mixup for reducing label noise in CLL.

**Surrogate Complementary Loss with ICM**   We propose a data augmentation for the existing loss-based CLL algorithms. In CLL, to minimize the non-convex $(0-1)$ loss in complementary learning, a common approach in statistical learning is to use a convex surrogate loss to approximate the target loss. In our work, we use $\ell$ to denote the *surrogate complementary loss (SCL)* loss functions and we combine our proposed data augmentation technique ICM with SCL during the training process. The main idea behind ICM data augmentation for complementary labels is to assign a new complementary label and incorporate additional information from samples within the same cluster, which share the same ordinary label. This approach enables the model to access not only more complementary labels but also new information about the samples.

---

[2]Intra-Class Mixup Noise-Free: is a proof-of-concept variant we designed prior to our proposed method. In this setting, Mixup is applied only between samples from the same class, so no additional label noise is introduced.
[3]Extra-Class Mixup Noise-Free: Mixup is applied between samples cross different class.

---

**Algorithm 1:** ICM training with cluster-consistent Mixup. Lines 1–3: extract SimSiam embeddings and assign $k$ clusters. Lines 4–12: synthesize $(\tilde{\mathbf{x}}, \tilde{y})$ by interpolating pairs within the same cluster using Eq. (6)–(7). Lines 13–14: update $\theta$ on the synthetic batch.

---

**Input:** Complementary-labeled dataset $\bar{\mathcal{D}} = \{(\mathbf{x}_i, \bar{y}_i)\}_{i=1}^{N}$, model $f_\theta$.
**Output:** Trained parameters $\theta$.

**1** **(1)** Embedding: $\mathbf{z}_i \leftarrow \mathcal{F}_{\text{sim}}(\mathbf{x}_i)$ for $i = 1, \ldots, N$      // $\mathcal{F}_{\text{sim}}$: `pretrained SimSiam encoder`

**2** **(2)** Clustering: run $k$-means on $\{\mathbf{z}_i\}$ to obtain cluster labels $c_i \in \{1, \ldots, k\}$.

**3** **(3)** Augment data: $\bar{\mathcal{D}} \leftarrow \{(\mathbf{x}_i, \bar{y}_i, c_i)\}_{i=1}^{N}$.

**4** **while** *not converged* **do**

**5**    **(4)** Sample a minibatch $\mathcal{B} \subset \bar{\mathcal{D}}$.

**6**    **(5)** For each cluster $u \in \{1, \ldots, k\}$, form $\mathcal{B}_u = \{(\mathbf{x}, \bar{y}, c) \in \mathcal{B} : c = u\}$.

**7**    **(6)** Initialize synthetic set $\tilde{\mathcal{B}} \leftarrow \emptyset$.

**8**    **foreach** $u$ *with* $|\mathcal{B}_u| \geq 2$ **do**

**9**      **for** $m = 1$ **to** $M_u$ **do**

**10**        (a) Draw two distinct pairs $(\mathbf{x}_i, \bar{y}_i, u), (\mathbf{x}_j, \bar{y}_j, u) \in \mathcal{B}_u$. // `sampling the samples in same cluster.`

**11**        (b) Sample $\lambda \sim \text{Beta}(\alpha, \alpha)$.

**12**        (c) Obtain ICM input $\tilde{\mathbf{x}}$ using Eq. (6).

**13**        (d) Compute label mixing coefficient $\lambda_{\bar{y}}$.

**14**        (e) Generate ICM label $\tilde{y}$ using Eq. (7).

**15**        (f) Append $(\tilde{\mathbf{x}}, \tilde{y})$ to $\tilde{\mathcal{B}}$.

**16**    **(7)** Compute loss: $\mathcal{L}(\theta) \leftarrow \frac{1}{|\tilde{\mathcal{B}}|} \sum_{(\tilde{\mathbf{x}}, \tilde{y}) \in \tilde{\mathcal{B}}} \mathcal{L}\big(g(\tilde{\mathbf{x}}), \tilde{y}\big)$.      // `training model with new synthetic data.`

**17**    **(8)** Update: $\theta \leftarrow \theta - \eta \nabla_\theta \mathcal{L}(\theta)$.

---

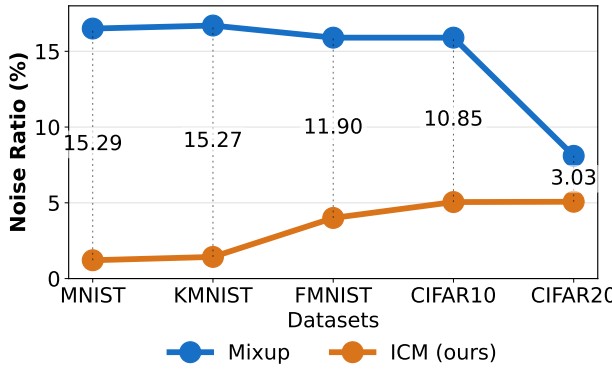

(a) Noise ratios of the Mixup and ICM methods across five datasets.

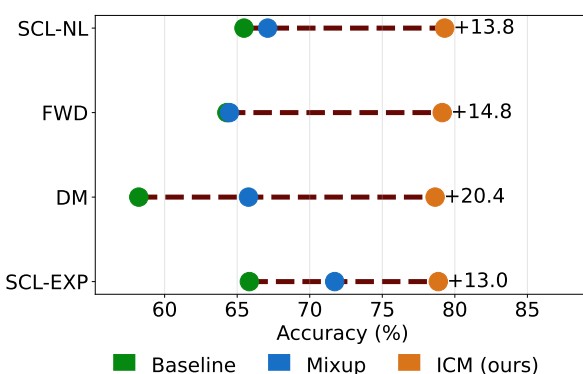

(b) Test accuracy of Mixup and ICM across different algorithms on CIFAR10

Figure 3: Comparison of the noise ratio across datasets (**left**) and the test accuracy of Mixup and ICM for different algorithms on CIFAR10 (**right**).

Additionally, it helps to reduce the noise that may arise when selecting pairs for data augmentation during training, thereby improving the overall learning process.

In loss-based complementary learning algorithms, a loss function $\ell : [K] \times \mathbb{R}^K \to \mathbb{R}$ is employed, which takes as input both the complementary label $\bar{y}_i$ and the prediction output of the model $g(\mathbf{x}_i)$. The objective of learning process is to minimize this loss function $\ell$ over the complementary dataset $\bar{D}$, which can be formulated as:

$$\mathcal{L}(g; \ell) = \frac{1}{N} \sum_{i=1}^{N} \ell\big(\bar{y}_i, g(\mathbf{x}_i)\big). \tag{8}$$

When incorporating the ICM data augmentation during training, the CLL loss function is updated as follows:

$$\mathcal{L}'(g;\ell) = \frac{1}{N}\sum_{i=1}^{N}\ell\Big(\tilde{y}_{i,j}, g(\tilde{\mathbf{x}}_{i,j})\Big) = \frac{1}{N}\sum_{i=1}^{N}\Big[\lambda\ell\Big(\bar{y}_i, g(\tilde{\mathbf{x}}_{i,j})\Big) + (1-\lambda)\ell\Big(\bar{y}_j, g(\tilde{\mathbf{x}}_{i,j})\Big)\Big], \tag{9}$$

where $\lambda \in [0,1] \sim \beta(\alpha,\alpha)$ (*beta distribution*), for $\alpha \in (0,\infty)$, $\tilde{\mathbf{x}}_{i,j}$ in equation 6, $\tilde{y}_{i,j}$ in equation 7, $j$ is random sampling from the same cluster of $i$, and $N$ is the size of training dataset. To better distinguish from $0-1$ based methods, we use a convex surrogate loss to approximate the target loss, denotes $\ell$. In fact, previous research in complementary learning has revealed similar patterns focused on minimizing the predictions of label classes, including approaches such as:

- Negative learning loss (SCL-NL) in (Kim et al., 2019) a modified log loss specifically designed for negative learning with complementary labels:

$$\ell_{\text{NL}}\big(\bar{y}, g(\mathbf{x})\big) = -\log(1 - \mathbf{p}_{\bar{y}} + \gamma), \text{where } 0 < \gamma < 1. \tag{10}$$

- Exponential loss (SCL-EXP) (Chou et al., 2020b):

$$\ell_{\text{EXP}}\big(\bar{y}, g(\mathbf{x})\big) = \exp(\mathbf{p}_{\bar{y}}). \tag{11}$$

- Forward correction (FWD) in (Yu et al., 2018) is a method for correcting loss using a forward correction approach based on a given transition matrix T:

$$\ell_{\text{FWD}}\big(\bar{y}, g(\mathbf{x})\big) = \ell(\bar{y}, T^T \mathbf{p}). \tag{12}$$

Here, $\gamma$ is a constant added to the loss function to prevent the SCL-NL loss from approaching infinity when $\mathbf{p}_{\bar{y}}$ equals 1, $\mathbf{p} \in \Delta^{K-1}$ represents the probability output of learning model if $g$ passes through a softmax layer, and $\Delta^{K-1}$ is the $K$-dimensional simplex.

## 4 Experiments

In this section, we evaluate ICM on synthetic and real-world datasets under both balanced and imbalanced conditions, comparing it with state-of-the-art CLL baselines. Our findings demonstrate that ICM significantly enhances performance and effectively addresses key CLL challenges.

### 4.1 Experiment Setup

**Datasets.** We assess the effectiveness of our proposed ICM framework across five synthetic labeled datasets: CIFAR10, CIFAR20, MNIST, KMNIST, and FMNIST. The synthetic labeled datasets consist of CIFAR10 (Krizhevsky et al., 2009) and CIFAR20, each containing 50,000 training samples and 10,000 testing samples. CIFAR10 encompasses 10 classes, whereas CIFAR20 comprises 20 superclasses derived from CIFAR100 (Krizhevsky et al., 2009). We do not benchmark on CIFAR100, as existing CLL algorithms have not demonstrated the ability to learn a meaningful classifier on this dataset when given only one complementary label per data instance. MNIST (Lecun et al., 1998), KMNIST (Clanuwat et al., 2018), and FMNIST (Xiao et al., 2017) each consist of 60,000 training samples and 10,000 testing samples, with all three datasets featuring ten classes.

Additionally, we evaluate our framework on real-world labeled datasets, including CLCIFAR10 and CLCIFAR20 (Wang et al., 2025), which use the images from CIFAR10 and CIFAR20, respectively, with complementary labels annotated by humans. In CLL, MNIST and CIFAR are standard datasets. Researchers have not transitioned to large-scale datasets with numerous classes, such as TinyImageNet (Le & Yang, 2015) and ImageNet (Deng et al., 2009). Preliminary tests reveal that state-of-the-art CLL algorithms struggle to produce meaningful classifiers for 100 classes, even with uniformly and noiselessly generated synthetic complementary labels. This is why existing CLL algorithms are evaluated on datasets with 10, 20 classes.

Table 1: Top-1 validation accuracy (%) on balanced (*bal*) $\rho = 1$ and long-tailed imbalanced (*imb*) ratio $\rho = 10$, $K$ cluster = 50 setups. The methods used are *SCL-NL* (S-NL), *FWD-INT* (FWD), *SCL-EXP* (S-EXP), *DM* losses, and ResNet18. Best performance is *highlighted in bold*.

| Method | Imbalanced *(imb)* | | Balanced *(bal)* | |
|---|---|---|---|---|
| | CLCIFAR10 | CLCIFAR20 | CLCIFAR10 | CLCIFAR20 |
| S-NL | $17.77_{0.20}$ | $5.80_{0.03}$ | $37.59_{0.40}$ | $8.53_{0.24}$ |
| S-NL+Mix | $21.28_{0.51}$ | $6.64_{0.48}$ | $42.96_{0.54}$ | $9.13_{0.44}$ |
| S-NL+ICM (ours) | $\mathbf{28.44}_{0.05}$ | $\mathbf{7.55}_{0.08}$ | $\mathbf{56.63}_{0.61}$ | $\mathbf{11.26}_{0.24}$ |
| DM | $15.19_{0.15}$ | $5.76_{0.06}$ | $38.20_{0.68}$ | $8.34_{0.08}$ |
| DM+Mix | $22.99_{0.22}$ | $6.92_{0.21}$ | $42.61_{0.48}$ | $9.12_{0.32}$ |
| DM+ICM (ours) | $\mathbf{27.88}_{0.94}$ | $\mathbf{7.10}_{0.02}$ | $\mathbf{53.04}_{0.40}$ | $\mathbf{11.47}_{0.18}$ |
| FWD | $12.07_{0.01}$ | $5.98_{0.17}$ | $42.98_{0.36}$ | $21.10_{0.23}$ |
| FWD+Mix | $17.06_{0.89}$ | $6.10_{0.16}$ | $42.38_{0.05}$ | $21.48_{0.19}$ |
| FWD+ICM (ours) | $\mathbf{18.23}_{0.08}$ | $\mathbf{7.73}_{0.09}$ | $\mathbf{58.97}_{0.21}$ | $\mathbf{35.94}_{0.33}$ |
| S-EXP | $17.37_{0.16}$ | $5.99_{0.21}$ | $41.42_{0.68}$ | $8.56_{0.25}$ |
| S-EXP+Mix | $20.38_{0.40}$ | $6.84_{0.13}$ | $43.56_{0.13}$ | $9.04_{0.21}$ |
| S-EXP+ICM (ours) | $\mathbf{27.52}_{0.06}$ | $\mathbf{7.01}_{0.08}$ | $\mathbf{56.26}_{0.15}$ | $\mathbf{11.20}_{0.06}$ |

**Baseline Methods.** Our framework can be applied with different methods, we choose SCL-EXP, SCL-NL (Chou et al., 2020b), FWD-INT (Yu et al., 2018), DM (Gao & Zhang, 2021) as our cooperators to validate the efficacy of our approach.

**Implementation Details.** For a fair comparison, we choose RestNet18 (He et al., 2016) as our backbone. We train the model with a batch size 512 for 300 epochs and an initial learning rate of $10^{-4}$, weight decay $10^{-4}$, and optimizer Adam (Ye et al., 2024). In the long-tailed imbalance setting (Cui et al., 2019; Cao et al., 2019), the difficulty of a dataset is commonly characterized by the *class imbalance ratio*, defined as $\rho = \frac{\max_i n_i}{\min_i n_i}$, where $n_i$ denotes the number of samples in class $i$. A dataset is said to exhibit *long-tailed imbalance* with ratio $\rho$ when the class sizes follow an exponentially decreasing sequence whose common ratio is $\rho^{1/(K-1)}$ across the $K$ classes. This construction ensures that the ratio between the largest (head) class and the smallest (tail) class is exactly $\rho$. All the experiments were run with Tesla V100-SXM2, 32GB memories. The hyper-parameters can be appropriately tuned via the validation process. For each subtask, we run the experiments three times. Other implemented details, including hyper-parameter selection through validation process such as $\alpha$, K cluster, can be found in our supplementary materials Appendix D. We experiment our proposed method across a wide spectrum of both balanced and imbalanced CLL settings. For imbalanced CLL, we follow (Cao et al., 2019) to generate a long-tailed distribution dataset with different imbalance ratios on ordinary datasets. Details of the different imbalanced setups can be found in the Appendix B.

## 4.2 Results and Analysis

We compare our results with several baselines, including *without Mixup* and *with Mixup (Mix)*, to verify the efficacy of our proposed method. Additionally, our method can be integrated with various base algorithms such as SCL-NL, FWD-INT, DM, and SCL-EXP.

The results for the real-world labeled datasets, CLCIFAR10 and CLCIFAR20, both in balanced and imbalanced settings with different loss functions, are presented in Table 1. For the synthetic labeled datasets (CIFAR10, CIFAR20, MNIST, KMNIST, and FMNIST) with *setup 1*, spanning from balanced to various imbalance ratios, detailed results are shown in Tables 2. Additional experimental details for *setup 2* and *setup 3*, with varying imbalanced ratios, can be found in Appendix B. Our proposed method consistently outperforms the baselines across all setups, from balanced to imbalanced scenarios, and achieves significant performance improvements when integrated with different base algorithms. To assess whether the observed improvements are statistically significant, we compute *p*-values on the CIFAR-10 and CLCIFAR-10 datasets.

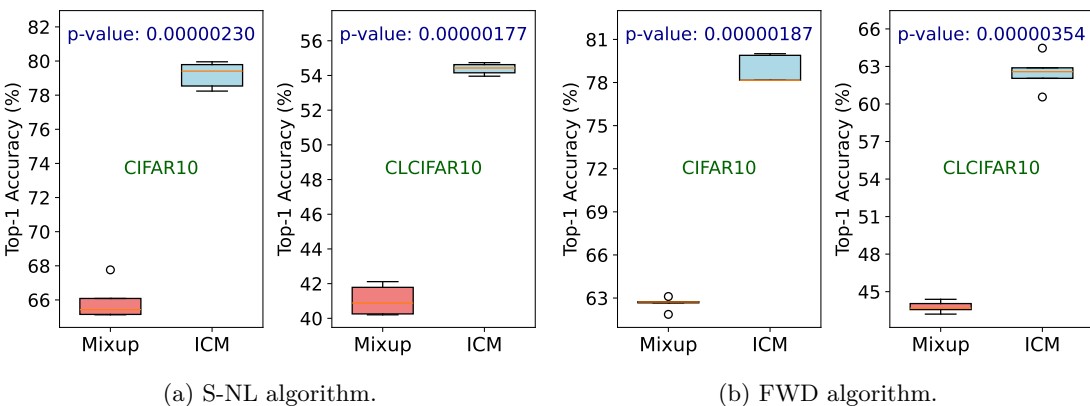

(a) S-NL algorithm.  (b) FWD algorithm.

Figure 4: Comparing the p-value of different between Mixup and ICM method on CIFAR10 and CLCIFAR10 with S-NL (**right**) and FWD (**left**) algorithms on both balanced and imbalanced ($\rho = 100$) scenarios.

Table 2: Top-1 validation accuracy (%) for $\rho = 100$ (long-tailed imbalanced) and $\rho = 1$ (balanced) setups across CIFAR10, CIFAR20, MNIST, KMNIST, and FMNIST datasets using ResNet18 and different loss methods.

| Method | CIFAR10 | | CIFAR20 | | MNIST | | KMNIST | | FMNIST | |
|---|---|---|---|---|---|---|---|---|---|---|
| | $\rho = 100$ | $\rho = 1$ | $\rho = 100$ | $\rho = 1$ | $\rho = 100$ | $\rho = 1$ | $\rho = 100$ | $\rho = 1$ | $\rho = 100$ | $\rho = 1$ |
| S-NL | $22.41_{0.31}$ | $65.47_{0.05}$ | $10.65_{0.28}$ | $24.14_{0.33}$ | $50.15_{0.52}$ | $97.78_{0.21}$ | $35.17_{0.17}$ | $88.92_{0.14}$ | $51.93_{0.33}$ | $85.15_{0.16}$ |
| S-NL+Mix | $31.46_{0.59}$ | $67.10_{0.11}$ | $13.47_{0.46}$ | $26.45_{0.07}$ | $54.23_{0.48}$ | $96.64_{0.11}$ | $37.25_{0.53}$ | $79.04_{0.16}$ | $56.07_{0.57}$ | $84.35_{0.08}$ |
| S-NL+ICM | $\mathbf{36.21}_{0.19}$ | $\mathbf{79.13}_{0.04}$ | $\mathbf{18.11}_{0.31}$ | $\mathbf{39.17}_{0.15}$ | $\mathbf{85.83}_{0.19}$ | $\mathbf{98.20}_{0.10}$ | $\mathbf{63.70}_{0.34}$ | $\mathbf{89.09}_{0.08}$ | $\mathbf{67.80}_{0.40}$ | $\mathbf{85.25}_{0.99}$ |
| FWD | $22.20_{0.40}$ | $64.29_{0.33}$ | $8.43_{0.29}$ | $23.18_{0.34}$ | $50.26_{0.57}$ | $97.49_{0.08}$ | $35.29_{0.37}$ | $80.41_{0.36}$ | $51.89_{0.63}$ | $84.16_{0.07}$ |
| FWD+Mix | $29.03_{0.39}$ | $64.47_{0.26}$ | $14.55_{0.46}$ | $22.79_{0.06}$ | $52.44_{0.49}$ | $94.09_{0.37}$ | $37.77_{0.47}$ | $70.83_{0.11}$ | $53.24_{0.48}$ | $82.43_{0.59}$ |
| FWD+ICM | $\mathbf{39.71}_{0.29}$ | $\mathbf{79.22}_{0.03}$ | $\mathbf{21.76}_{0.26}$ | $\mathbf{42.20}_{0.09}$ | $\mathbf{85.27}_{0.69}$ | $\mathbf{98.18}_{0.10}$ | $\mathbf{63.50}_{0.47}$ | $\mathbf{88.92}_{0.04}$ | $\mathbf{67.90}_{0.47}$ | $\mathbf{84.75}_{0.11}$ |
| DM | $20.91_{0.15}$ | $58.22_{0.24}$ | $10.16_{0.05}$ | $21.43_{0.09}$ | $51.28_{0.33}$ | $95.10_{0.05}$ | $32.60_{0.20}$ | $73.98_{0.25}$ | $49.46_{0.08}$ | $82.68_{0.34}$ |
| DM+Mix | $30.52_{0.31}$ | $65.78_{0.22}$ | $13.27_{0.12}$ | $24.95_{0.66}$ | $55.78_{2.36}$ | $95.69_{0.10}$ | $36.37_{1.19}$ | $78.15_{0.35}$ | $54.22_{0.56}$ | $82.77_{0.37}$ |
| DM+ICM | $\mathbf{36.37}_{0.17}$ | $\mathbf{78.64}_{0.05}$ | $\mathbf{17.42}_{0.52}$ | $\mathbf{38.48}_{0.13}$ | $\mathbf{85.91}_{0.15}$ | $\mathbf{98.67}_{0.07}$ | $\mathbf{66.98}_{4.79}$ | $\mathbf{89.60}_{0.52}$ | $\mathbf{66.46}_{1.19}$ | $\mathbf{84.61}_{0.24}$ |
| S-EXP | $22.96_{0.24}$ | $65.84_{0.23}$ | $10.13_{0.29}$ | $24.07_{0.10}$ | $50.29_{0.06}$ | $98.68_{0.21}$ | $35.62_{0.05}$ | $90.38_{0.08}$ | $51.30_{0.10}$ | $85.23_{0.09}$ |
| S-EXP+Mix | $31.51_{0.18}$ | $71.72_{0.29}$ | $13.77_{0.17}$ | $27.90_{0.25}$ | $53.68_{0.28}$ | $98.27_{0.12}$ | $37.52_{1.02}$ | $88.40_{0.23}$ | $53.50_{3.53}$ | $84.42_{0.21}$ |
| S-EXP+ICM | $\mathbf{36.70}_{0.10}$ | $\mathbf{78.86}_{0.12}$ | $\mathbf{16.75}_{0.15}$ | $\mathbf{38.91}_{0.22}$ | $\mathbf{86.06}_{0.37}$ | $\mathbf{98.81}_{0.03}$ | $\mathbf{64.46}_{0.04}$ | $\mathbf{90.45}_{0.12}$ | $\mathbf{64.03}_{3.31}$ | $\mathbf{85.73}_{0.20}$ |

The results show that the *p*-values are well below 0.001, providing strong evidence that our proposed method significantly outperforms the original Mixup. These results are summarized in Figure 4 and reported in more detail in Appendix D.5.

Moreover, we conduct another analyses demonstrating that our proposed method, ICM, proves to be a competitive approach for enhancing CLL. This is evidenced by our assessment of the enhancing robustness of ICM when combining with various data augmentation techniques, ranging from weak (Flipflop, Cutout (De-Vries & Taylor, 2017)) to strong (AutoAug (Cubuk et al., 2019), RandAug (Cubuk et al., 2020)). The results in Figure 5a illustrate the significant benefits of combining ICM with various data augmentation techniques, for instance, on the CIFAR10 dataset, the combination of ICM with these augmentations achieves accuracy levels approaching 80%, far surpassing the results of their counterparts without ICM. Interestingly, Cutout appears to hurt performance when used together with ICM. A plausible explanation is that applying ICM on top of Cutout may excessively remove informative regions of the input, leading to overly distorted synthetic samples.

Furthermore, we conduct a series of analyses demonstrating that our proposed method, ICM, proves to be a competitive approach for enhancing CLL. This is evidenced by our assessment of the *bias* and *variance* of the empirical *Gradient Analysis* in next section. It is also crucial to highlight that the benefits of sharing new synthetic data extend beyond merely sharing complementary labels in the CLL context. This assertion is supported by an ablation study where we share *new data*, *soft label*, and *hard label* during the model

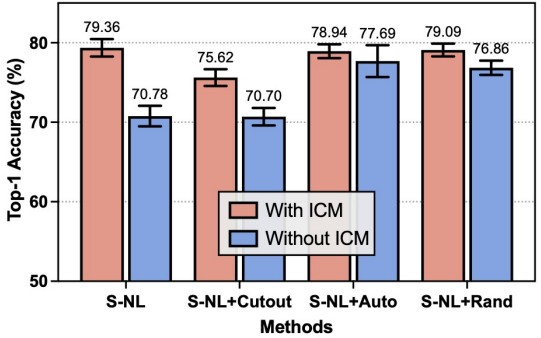

(a) Enhancing robustness by combining ICM with weak and strong data augmentation techniques on CIFAR10.

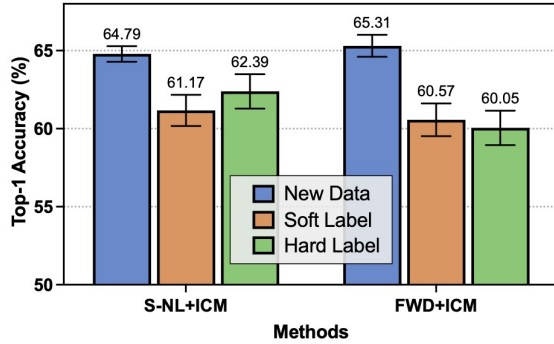

(b) Comparison of new, soft, and hard label sharing strategies under S-NL and FWD losses on CIFAR10.

Figure 5: Experimental results on CIFAR-10: (a) robustness gains from combining ICM with weak and strong data augmentations; (b) performance comparison of new, soft, and hard label-sharing strategies under S-NL and FWD losses.

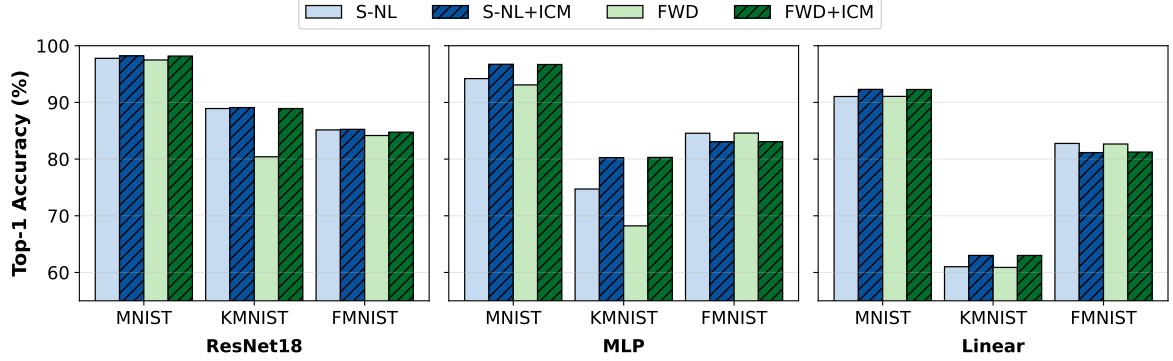

Figure 6: Comparing different model architecture with ICM method on MNIST family dataset under balanced setup.

training process. The detailed results presented in Figure 5b. In addition, we investigate methods for mitigating class imbalance in CLL. We introduce *Multi Intra-Cluster Mixup (MICM)*, which extends intra-cluster mixing to generate synthetic samples under imbalanced class distributions, thereby encouraging more effective complementary-label sharing for minority classes. Technical details of MICM and additional empirical results are provided in Appendices C and E.

Additionally, we evaluate the effectiveness of our proposed method (ICM) across models of varying complexity, including linear classifiers, multilayer perceptrons (MLPs), and ResNet18, to examine how architectural capacity interacts with ICM. Our empirical results show that ICM consistently improves performance with ResNet18 on all three datasets (MNIST, KMNIST, and FMNIST). For MLPs and linear models, ICM yields clear gains on MNIST and KMNIST, but leads to degraded performance on FMNIST. Detailed results are reported in Figure 6 and Appendix D.6.

Taken together, these analyses motivate a broader perspective on the practicality of CLL. In recent years, the field has observed that CLL is still not fully practical, especially when moving from synthetic to real-world datasets and increasing the number of classes. We found that the current state-of-the-art algorithms struggle under these conditions, highlighting the need for further work to make CLL more applicable in practice (Wang et al., 2025; Ye et al., 2024). Our proposed method, ICM, introduces a novel data augmentation approach that aims to make CLL more realistic. Specifically, ICM is designed to mitigate the effects of complementary-label noise associated with synthetic complementary samples. Through rigorous empirical evaluation, ICM demonstrates the effectiveness of encouraging complementary-label sharing among nearby examples, leading

to consistent performance improvements across a wide range of experimental setups. Our empirical results further show that ICM substantially improves the performance of learning models across various state-of-the-art algorithms. In particular, when we applied ICM to a real-world dataset, CLCIFAR10, the model performance increased by 10%. We hope that our work helps practitioners develop more accurate and reliable models in real-world scenarios characterized by complementary-label learning.

### 4.3 Gradient Analysis

We further discuss how the ICM framework gives such improvement by arranging the learning process via gradient analysis. This discussion centers on examining loss gradients within the experimental setup, particularly the *stochastic gradient (SGD)* employed in mini-batch optimization. Specifically, we evaluate the bias-variance tradeoff of the gradient estimation error involving complementary gradients with ICM and the Mixup method versus the ordinary gradient. To provide a more accurate assessment, we utilize the bias-variance decomposition technique. Traditionally used in statistical learning to assess algorithmic complexity, we extend this framework to evaluate the estimation error of the gradient, setting the ordinary gradient as the target. We will show that our proposed framework ICM has a lower *mean squared error (MSE)* than the original Mixup, caused by its slight variance and bias.

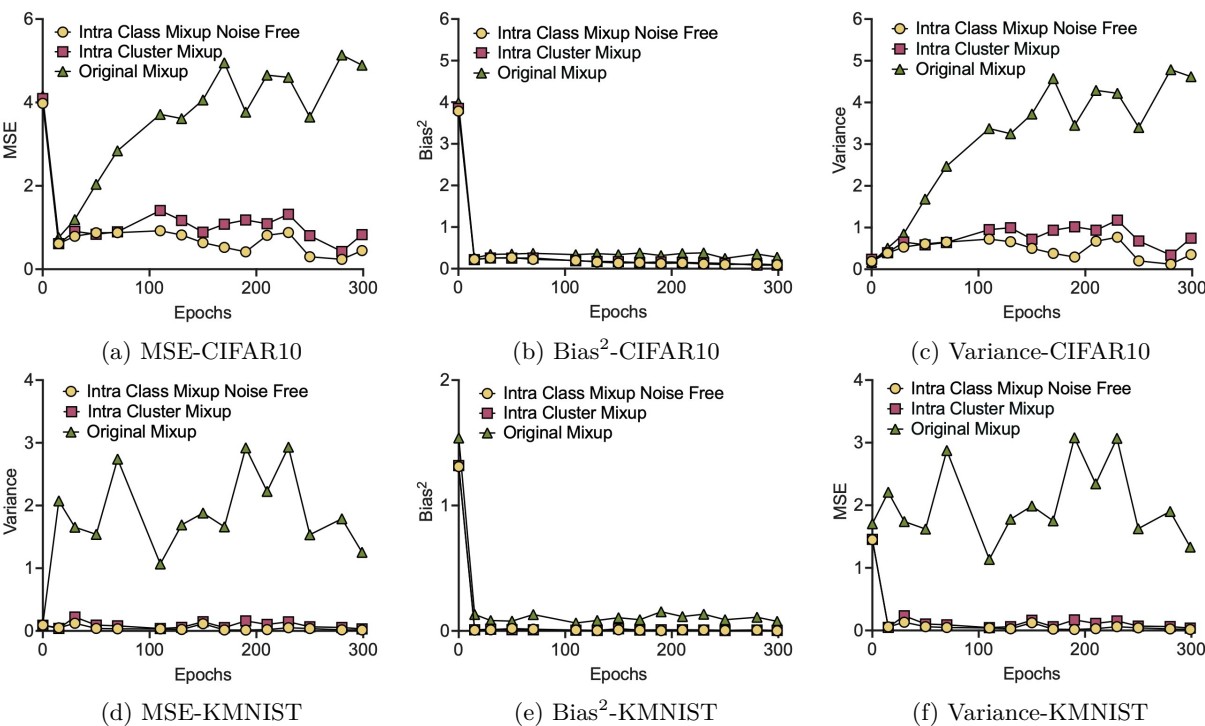

Figure 7: Comparison of gradient estimation errors between original Mixup and *Mixup Noise-Free* on MNIST and CIFAR10, using the SCL-NL loss function and ResNet18 architecture. *Mixup Noise-Free* demonstrates lower gradient estimation error than the original Mixup on both datasets, attributed to reduced noise interference, which impacts classifier performance in CLL contexts.

We represent the gradient step determined by ordinary labeled data $(\mathbf{x}, y)$ and ordinary loss $\ell$ as $f$. The complementary gradient step, considering complementary labeled data $(\mathbf{x}, \bar{y})$ and complementary loss $\bar{\ell}$ or $(\phi)$, is denoted as $c$. Additionally, $b$ denotes the expected gradient step of $[K]\backslash\{y\}$, calculated as the average of $c$ across all possible complementary labels. This can be formalized as follows:

$$f = \nabla\ell\big(y, g(\mathbf{x})\big), \tag{13}$$

$$c = \nabla\bar{\ell}\big(\bar{y}, g(\mathbf{x})\big), \tag{14}$$

$$b = \frac{1}{K-1} \sum_{y' \neq \bar{y}} \nabla \bar{\ell}\big(y', g(\mathbf{x})\big). \tag{15}$$

We designate $f$ as the ground truth, representing the target complementary estimator $c$. We expect the MSE of the gradient estimation to be minimal.

$$\text{MSE} = \mathbb{E}_{\mathbf{x}, y, \bar{y}}[(f - c)^2]. \tag{16}$$

We drive the bias-variance decomposition by introducing $b$ and eliminating remaining terms:

$$\mathbb{E}\big[(f - c)^2\big] = \mathbb{E}[(f - b + b - c)^2], \tag{17}$$

$$= \underbrace{\mathbb{E}[(f - b)^2]}_{\texttt{Bias}^2} + \underbrace{\mathbb{E}[(b - c)^2]}_{\texttt{Variance}}. \tag{18}$$

We conduct experiments to assess how well the complementary gradient $c$ approximates the ordinary gradient $f$ and compare it with a baseline method (*original Mixup*). The training process is as follows:

In each epoch, we compute three gradients, namely the ordinary gradient $f$, the current method $c$, and $b$. We evaluate the MSE, the square bias term, and the variance term using equation 16 and equation 18. In each epoch, we update the model only with $f$ to ensure a fair comparison of gradients. The optimizer used was SGD with a learning rate of $10^{-4}$, and the training was conducted for 300 epochs.

The results presented in Figure 7 indicate that Mixup exhibits a higher MSE due to elevated levels of variance and bias. Conversely, ICM demonstrates significantly lower variance and bias when compared to Mixup, aligning more closely with the ideal case of *Intra Class Mixup Noise Free*. This supports our observation that our proposed method outperforms Mixup in CLL context by achieving lower variance and bias.

## 5 Conclusion

This paper presented a novel data augmentation approach, ICM, specifically designs to mitigate the effects of *complementary-label noise* associated with synthetic complementary samples by synthesizing augmented data only within the same cluster. Through rigorous empirical evaluations across diverse CLL settings, we have demonstrated the effectiveness of encouraging complementary label sharing of nearby examples, leading to consistent performance improvements across a wide spectrum of experimental setups, from synthetic to real-world labeled datasets, in both balanced and imbalanced CLL settings. Our empirical experiments reveal that ICM substantially enhances the performance of learning models across a variety of state-of-the-art algorithms. Additionally, our investigations highlight the heightened sensitivity of classifiers trained under CLL conditions to *complementary-label noise*, which leads to performance degradation of CLL models. These findings underscore the significant contribution of ICM to the field of CLL. By providing a data augmentation strategy that effectively tackles the issue of *complementary-label noise*, ICM empowers practitioners to develop more accurate and reliable models in real-world scenarios characterized by CLL.

## 6 Limitation and Future Works

Despite its contributions, this study has several limitations. First, when applied to simpler models such as linear classifiers and multilayer perceptrons (MLPs) on benchmark FMIST datasets, our augmentation technique yields reduced performance. This decline stems from the limited capacity of these models, which struggle to accommodate the added complexity and increased overlap in feature representations introduced by ICM. Second, we have not yet evaluated our approach in scenarios where instances carry multiple complementary labels. Investigating the benefits of ICM in a multi-complementary-label learning setting remains an important direction for future work.

## Broader Impact Statement

We used publicly available benchmarks, including MNIST, KMNIST, FMNIST, CIFAR10, CIFAR20, CLCIFAR10, and CLCIFAR20, and identified no significant ethical concerns in their use. To tackle the limited

supervision inherent in these datasets, we optimized our learning algorithm to efficiently extract insights from both balanced and imbalanced class distributions. This approach is especially valuable in scientific settings where true labels are sensitive or costly to obtain. Moreover, by relying on complementary labels, we preserve data privacy and reduce annotation costs without sacrificing model accuracy.

## Acknowledgement

We thank the anonymous reviewers and members of CLLab for their constructive feedback. This work is partially supported by the National Science and Technology Council in Taiwan via NSTC 113-2634-F-002-008, 114-2221-E-002-102-MY3, NTU AI Center of Research Excellence within Taiwan Centers of Excellence, and NTU Center for Data Intelligence via NTU-114L900901. We thank to National Center for High-performance Computing (NCHC) of National Applied Research Laboratories (NARLabs) in Taiwan for providing computational and storage resources. H.-T. Lin is honored to be supported by the Leap Fellowship of the Foundation for the Advancement of Outstanding Scholarship in Taiwan since 2025.

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

# Appendix

## A  Proof

*Proof.* By construction, each Mixup sample is given by $\tilde{\mathbf{x}}_{i,j} = \lambda\mathbf{x}_i + (1-\lambda)\mathbf{x}_j, \quad \tilde{y}_{i,j} = \lambda\bar{y}_i + (1-\lambda)\bar{y}_j$, where we view $\tilde{y}_{i,j}$ as a convex combination of the one-hot complementary labels $\bar{y}_i$ and $\bar{y}_j \in \{1,\ldots,K\}$. Under the zero-one loss, $\ell(\bar{y}, g(\mathbf{x})) = [\![\bar{y} \neq g(\mathbf{x})]\!]$, the complementary classification risk under Mixup can be written as

$$\mathcal{R}'(g;\ell) = \frac{1}{N}\sum_{i=1}^{N}\ell(\tilde{y}_{i,j}, g(\tilde{\mathbf{x}}_{i,j}))$$

$$= \frac{1}{N}\sum_{i=1}^{N}\left[\lambda\,\ell(\bar{y}_i, g(\tilde{\mathbf{x}}_{i,j})) + (1-\lambda)\,\ell(\bar{y}_j, g(\tilde{\mathbf{x}}_{i,j}))\right]$$

$$= \lambda\frac{1}{N}\sum_{i=1}^{N}[\![\bar{y}_i \neq g(\tilde{\mathbf{x}}_{i,j})]\!] + (1-\lambda)\frac{1}{N}\sum_{i=1}^{N}[\![\bar{y}_j \neq g(\tilde{\mathbf{x}}_{i,j})]\!].$$

We now decompose each indicator using $[\![A \neq B]\!] = 1 - [\![A = B]\!]$. For the first sum,

$$\frac{1}{N}\sum_{i=1}^{N}[\![\bar{y}_i \neq g(\tilde{\mathbf{x}}_{i,j})]\!] = \frac{1}{N}\sum_{i=1}^{N}\left[1 - [\![\bar{y}_i = g(\tilde{\mathbf{x}}_{i,j})]\!]\right]$$

$$= \mathbb{E}_{(\mathbf{x},\bar{y})\sim\bar{D}}[\![\bar{y}_i \neq g(\tilde{\mathbf{x}}_{i,j})]\!] - \varepsilon_i,$$

where $\varepsilon_i$ denotes the local noise error associated with $\bar{y}_i$ (equation 3). An analogous decomposition holds for the second sum, yielding the contribution $(1-\lambda)\varepsilon_j$ from the local noise associated with $\bar{y}_j$. Substituting these decompositions back into the expression for $\mathcal{R}'(g;\ell)$ yields

$$\mathcal{R}'(g;\ell) = \lambda\,\mathbb{E}_{(\mathbf{x},\bar{y})\sim\bar{D}}[\![\bar{y}_i \neq g(\tilde{\mathbf{x}}_{i,j})]\!] + (1-\lambda)\,\mathbb{E}_{(\mathbf{x},\bar{y})\sim\bar{D}}[\![\bar{y}_j \neq g(\tilde{\mathbf{x}}_{i,j})]\!] + \lambda\varepsilon_i + (1-\lambda)\varepsilon_j,$$

which is exactly the claimed decomposition in equation 5. $\qquad\square$

## B  Generating Imbalanced Complementary Labels

In this section, we specifically consider three situations as the primary causes of imbalanced complementary labels:

*Setup 1: Imbalanced ordinary with a uniform transition matrix*

In this configuration, the imbalance in the ordinary distribution dataset results in a corresponding imbalance in complementary labels. The ratio of imbalance in complementary labels significantly decreases compared to the ordinary distribution (e.g., from 100 to around 1.4) when generated using a uniform transition matrix. This imbalance generation is visually represented as *setup 1* in Figure 8a.

*Setup 2: Balanced ordinary with a biased transition matrix*

Here, a biased transition matrix arises from the imbalance in complementary labels, as depicted in (b) *setup 2* in Figure 8b, where complementarity probabilities vary across classes. The imbalance in complementary labels aligns with the observed imbalance ratio in the transition matrix (e.g., both being 10). This correlation underscores the direct influence of complementary label distribution on the imbalance characteristics of the transition matrix in this setting.

*Setup 3: Imbalanced both ordinary and biased transition matrix*

The imbalance in both ordinary and biased transition matrix compounds the challenge in CLL. This setup intensifies the bias in complementary labels compared to the previous setups. *setup 3* is particularly challenging for the model to find a good classifier $g$ due to the under-representation of minority classes. Figure 8c illustrates the generation of imbalanced complementary labels in *setup 3*.

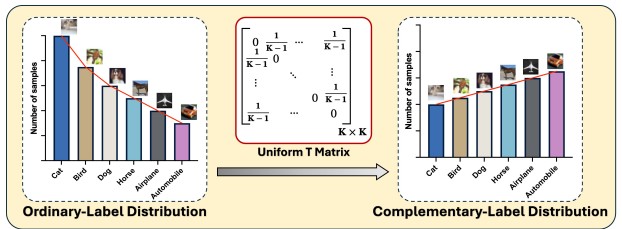

(a) Setup 1: Imbalanced ordinary with a uniform transition matrix.

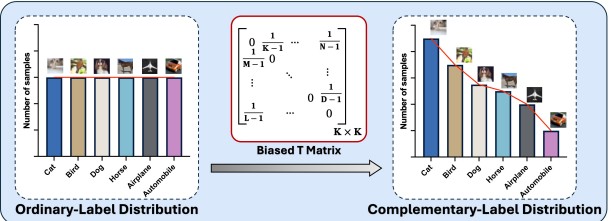

(b) Setup 2: Balanced ordinary with a biased transition matrix.

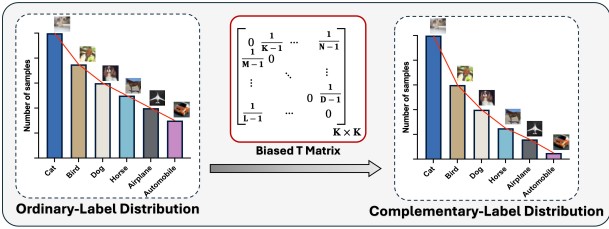

(c) Setup 3: Imbalanced both ordinary and biased transition matrix.

Figure 8: Illustration of the generation of imbalanced complementary label settings.

In our proposed setup, we introduce a setting of *long-tailed imbalance*, a concept seen in previous works (Cao et al., 2019; Cui et al., 2019). This setup is designed to generate both an imbalanced ordinary distribution and a biased transition matrix. The degree of *long-tailed imbalance* is characterized by the parameter $\rho$, which dictates class sizes through an exponentially decreasing sequence. The decreasing constant, represented as $\rho^{1/K-1}$, precisely controls the class imbalance ratio of $\rho$. To illustrate, consider an example of a long-tailed ordinary distribution in *setup 1*, visualized in Figure 8a.

## C  Imbalanced issue in Complementary-Label Learning

The most prevalent assumption in CLL is uniform generation, positing that complementary labels are generated from ordinary labels with equal probability (Cao et al., 2022; Feng et al., 2020; Lin & Lin, 2023; Ishida et al., 2018). Another research has delved into the non-uniform generation of complementary labels (Yu et al., 2018). However, regardless of whether the assumptions are uniform or non-uniform, existing benchmark datasets generally contain roughly balanced portions of complementary labels.

To the best of our knowledge, no one has explored *what strategies should be adopted when dealing with thousands of complementary labels that are highly imbalanced*? This gap in the literature motivates our study, which aims to explore and develop potential methodologies for addressing the issue of imbalances in CLL datasets. Imbalanced CLL may arise due to an uneven distribution in the ordinary dataset or an imbalanced transition matrix employed for generating complementary labels. In this study, we specifically consider three situations as the primary causes of imbalanced complementary labels: *setup 1:* Imbalance in ordinary with a uniform transition matrix, *setup 2:* Balanced ordinary with a biased transition matrix, and *setup 3:* Imbalance in both ordinary and a biased transition matrix. The illustration of 3 setups in detail is depicted in Appendix B.

In the Mixup, the mixing factor $\lambda$ remains the same for synthetic $\mathbf{x}$ and $\bar{y}$. However, a significant challenge in imbalanced classification is that minority classes are under-represented in the objective function, leading to poor generalization for these classes by the classifier (Cao et al., 2019; Ren et al., 2020). To address this issue, it is crucial to develop methods that enhance model learning for minority classes. Inspired by a successful Mixup variant that enhances class-imbalanced learning by making *y not uniform* (Chou et al., 2020a) and recognizing the essential need for sharing more complementary labels among minority classes,

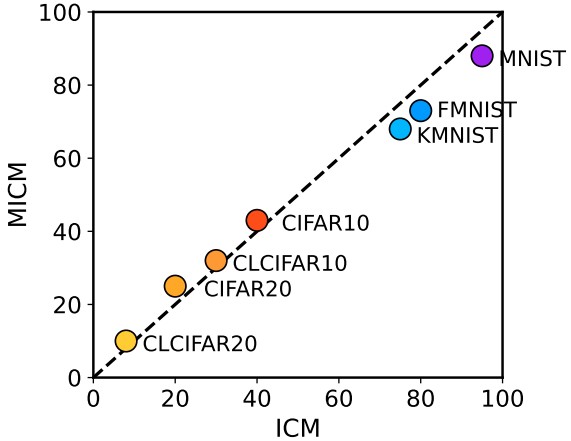
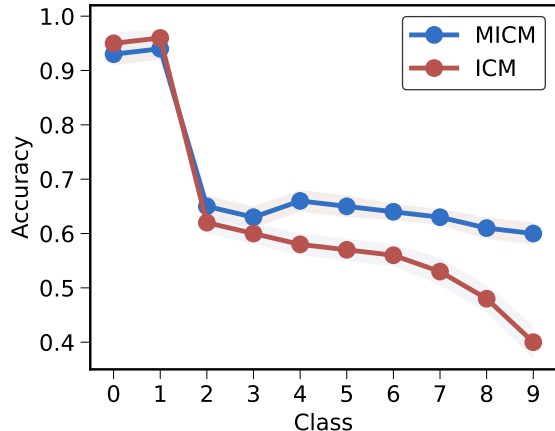

(a) Effectiveness of ICM vs MICM on different complementary datasets. The figure reveals that MICM outperforms on complex datasets like CIFAR and CLCIFAR, while ICM proves to be more efficient on simpler datasets, such as the MNIST family.

(b) Comparison of MICM vs ICM in imbalanced CLL on CIFAR10. This ablation study demonstrates that MICM not only achieves better overall performance in imbalanced CLL scenarios but also significantly enhances model learning for minority classes.

Figure 9: The illustration of ablation study on different setups.

we propose *Multi Intra-Cluster Mixup (MICM)*. MICM extends the mixing of samples within the same cluster when generating new synthetic data, thereby encouraging more complementary label sharing for minority classes. To verify this hypothesis, we conducted an ablation study comparing the effects of MICM versus ICM on enhancing the learning of minority classes in imbalanced CLL. This study was performed on setup 1, with an imbalance ratio $\rho = 10$, utilizing the SCL-NL loss function and the ResNet18 architecture. The results, as shown in Figure 9b, confirm that MICM improves the learning model in minority classes, thereby demonstrating its efficacy in imbalanced CLL. In MICM, we calculate $\lambda_{\bar{y}}$ based on *Inverse Distance Weighting (IDW)*. The core concept of IDW involves determining $\lambda_{\bar{y}}$ based on the distances among randomly selected samples in the cluster. For instance, when three samples are randomly chosen within a cluster, the new sample will be generated by equation 19, then this one serves as an anchor to calculate distances to the other three; the sample closer to the anchor is assigned a higher weight than the farther sample. Given three examples indexed by $i$, $j$, and $k$, MICM constructs a mixed feature vector and a mixed label as

$$\tilde{\mathbf{x}}_{i,j,k} = \lambda_1 \mathbf{x}_i + \lambda_2 \mathbf{x}_j + \lambda_3 \mathbf{x}_k, \tag{19}$$

$$\tilde{y}_{i,j,k} = \lambda_{\bar{y},i}\,\bar{y}_i + \lambda_{\bar{y},j}\,\bar{y}_j + \lambda_{\bar{y},k}\,\bar{y}_k. \tag{20}$$

The feature-mixing coefficients $(\lambda_1, \lambda_2, \lambda_3)$ are drawn from a Dirichlet distribution $(\lambda_1, \lambda_2, \lambda_3) \sim \text{Dir}(\alpha, \alpha, \alpha)$, so that $\lambda_r \in [0, 1]$ and $\sum_{r=1}^{3} \lambda_r = 1$. In contrast, the label-mixing coefficients $\lambda_{\bar{y},s}$ are computed adaptively based on the distances between $\tilde{\mathbf{x}}_{i,j,k}$ and each of the original samples:

$$\lambda_{\bar{y},s} = \frac{\dfrac{1}{d\big(\tilde{\mathbf{x}}_{i,j,k}, \mathbf{x}_s\big)}}{\displaystyle\sum_{t \in \{i,j,k\}} \frac{1}{d\big(\tilde{\mathbf{x}}_{i,j,k}, \mathbf{x}_t\big)}}, \quad s \in \{i, j, k\}, \tag{21}$$

$$d\big(\tilde{\mathbf{x}}_{i,j,k}, \mathbf{x}_s\big) = \begin{cases} C, & \text{if } \big\|\tilde{\mathbf{x}}_{i,j,k} - \mathbf{x}_s\big\|_2 = 0, \\ \big\|\tilde{\mathbf{x}}_{i,j,k} - \mathbf{x}_s\big\|_2, & \text{otherwise}, \end{cases} \tag{22}$$

where $d(\cdot, \cdot)$ denotes the Euclidean distance, $\|\cdot\|_2$ is the $\ell_2$ norm, and $C > 0$ is a small constant introduced to avoid division by zero. By construction, $\lambda_{\bar{y},s} \in [0, 1]$ and $\sum_{s \in \{i,j,k\}} \lambda_{\bar{y},s} = 1$.

Using Algorithm 1, our proposed MICM method obtains mixed input based on equation 19, mixed labels according to equation 20, and computes $\lambda_{\bar{y}}$ using equation 21, equation 22. The hyperparameter $C$ plays a crucial role; it is introduced to prevent a 0 value and offers flexibility in controlling the weight of $\lambda_{\bar{y}}$. Specifically, when $C$ is small, the weight of the anchor becomes more substantial, and conversely, when $C$ is large, the weight of the anchor diminishes. The fine-tuning of $C$ spectrum value is provided in *Parameterization* subsection.

## D  Additional Results of Ablation Study

### D.1  Comparing ICM vs. MICM in Imbalanced CLL

For imbalanced CLL, we follow (Cao et al., 2019) to generate a long-tailed distribution dataset with different imbalance ratios $\rho$ (10, 100) on ordinary datasets for *setup 1*. For *setup 2*, we use the same method as (Cao et al., 2019) to generate a biased transition matrix with imbalance ratios $\rho$ (3, 5, 10). In *setup 3*, we combine both long-tailed distributions on ordinary datasets with imbalance ratios $\rho$ (10, 50, 100) and a biased transition matrix with imbalance ratios $\rho$ (5, 10). The details of the experiment on five synthetic labeled datasets with these three setups and two real-world labeled datasets are presented in next subsection.

An intriguing observation is that while multiple Mixup augmentation may introduce significant noise (*cons*), its regularization effect (*pros*) may not be necessary for simpler tasks. This contrast makes the MICM method less effective than ICM on the MNIST family datasets.

However, for the more complex CIFAR datasets, the MICM method becomes preferable, as shown in Figure 9a, which compares the effectiveness of ICM and MICM on different complementary datasets. In imbalanced CLL scenarios, MICM also demonstrates superiority by more effectively enhancing learning in minority classes and improving overall model performance. Detailed results supporting this finding can be found in Figure 10b.

### D.2  Quantifying Model Performance: Mixup within the Same Complementary-Label vs. Intra-Cluster Mixup

To assess the model performance of Mixup within the same complementary-label and ICM, we conduct experiments on five datasets using *setup 1*. The configurations include a long-tailed imbalance ratio $\rho = 100$, $K = 50$, utilizing *SCL-NL* loss function, and the ResNet18 architecture. The results, presented in Figure 10a, clearly demonstrate a significant performance improvement with ICM compared to Mixup within the same complementary-label across all five datasets. These findings validate the superiority of our proposed method, ICM, over the Mixup within the same complementary-label approach.

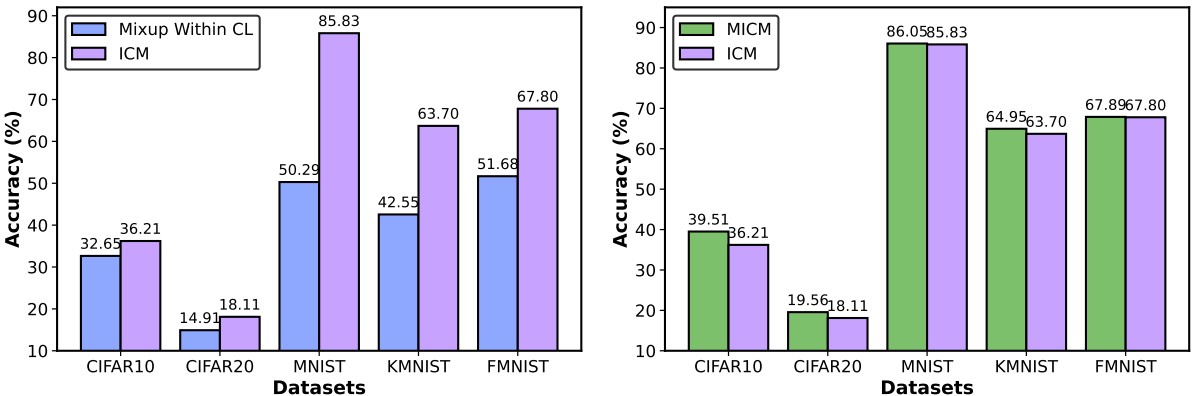

(a) Comparing ICM vs. Mixup within the same CL.    (b) Comparing ICM vs. MICM in imbalanced CLL.

Figure 10: Ablation study comparing our proposed ICM method with other Mixup variants using the S-NL method under Setup 1 with imbalance ratio $\rho = 100$ and ResNet18 architecture.

### D.3   Extra-Class Mixup Filter vs. Intra-Class Mixup Filter under Imbalanced CLL Setting

In this section, we perform an ablation study to compare the performance of models using Intra-Class Mixup Filter and Extra-Class Mixup Filter. This analysis aims to elucidate the advantages of Intra and Extra Mixup under the CLL scenario. We conduct experiments on CIFAR10 dataset using *setup 1*. The configurations include a long-tailed imbalance ratio $\rho = 100$, $K = 50$, SCL-NL loss, and the ResNet18 architecture. The results presented in Table 3 demonstrate a significant superiority of Intra-Class Mixup over Extra-Class Mixup in the CLL setting.

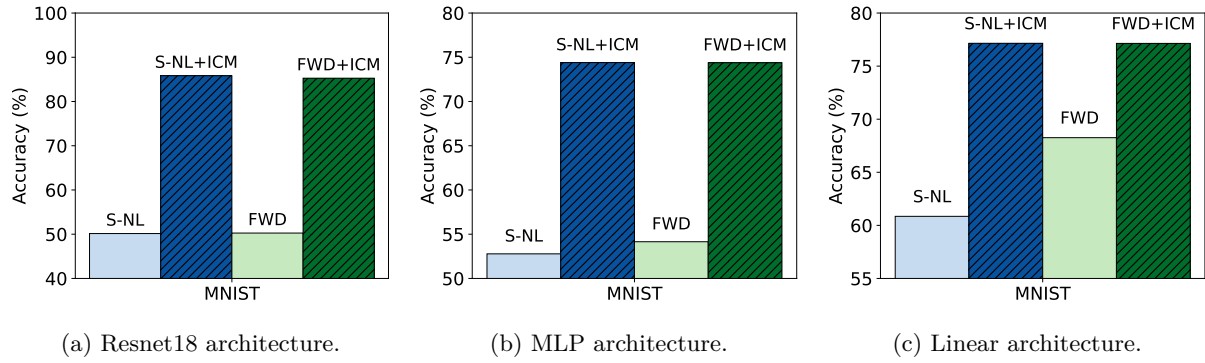

(a) Resnet18 architecture.   (b) MLP architecture.   (c) Linear architecture.

Figure 11: Comparing different model architectures and algorithms with ICM method on MNIST dataset under a long-tailed imbalanced ratio $\rho = 100$.

Table 3: Comparison of Extra-Class Mixup filter and Intra-Class Mixup filter in CLL setting with *setup 1* a long-tailed imbalance (*imb*) ratio $\rho = 100$, $K = 50$, SCL-NL loss, on CIFAR10 dataset, and ResNet18 architecture

|  | Noise Ratio | Accuracy |
|---|---|---|
| Extra-Class Mixup Filter | 0% | $37.17_{0.30}$ |
| Intra-Class Mixup Filter | 0% | $\mathbf{48.59}_{0.40}$ |

### D.4   Parameterization

#### D.4.1   Fine-tuning $\alpha$ Hyper-parameter

We conduct an ablation study to optimize the selection of the $\alpha$ parameter across five datasets: CIFAR10, CIFAR20, MNIST, KMNIST, and FMNIST. The study systematically explores a range from 0 to 2.0 to investigate the performance of different values of $\alpha$. The optimal $\alpha$ values, identified based on the outcomes of this study, are summarized in Table 4. Notably, the chosen $\alpha$ values for MNIST and KMNIST remain consistent across both the beta and dirichlet distributions, while those for CIFAR10, CIFAR20, and FMNIST varied between the two distributions.

Table 4: Selected $\alpha$ for beta distribution $\beta(\alpha, \alpha)$ and dirichlet distribution $Dir(\alpha, \alpha, \alpha)$

| Dataset | $\beta(\alpha, \alpha)$ | $Dir(\alpha, \alpha, \alpha)$ |
|---|---|---|
| CIFAR10 | 0.4 | 0.2 |
| CIFAR20 | 0.1 | 0.4 |
| MNIST | 0.1 | 0.1 |
| KMNIST | 0.3 | 0.3 |
| FMNIST | 0.1 | 0.4 |

### D.4.2   Fine-tuning K Cluster Hyper-parameter

Another ablation study aims to optimize the selection of the $K$ cluster hyper-parameter across the same five datasets. This study focuses on the *S-NL+ICM* method under *setup 1* with an imbalance ratio $\rho = 100$ and ResNet 18 architecture. The exploration spans a range from 30 to 90, and the results, presented in Figure 12, lead to the selection of $K = 50$ for all five datasets. This choice balances performance metrics and computational efficiency, offering reasonable performance and practicality in terms of training time.

### D.4.3   Fine-tuning C Constant Value

We conduct an ablation study to optimize the selection of the $C$ value across the same five datasets. This study focuses on the *S-NL+MICM* method under *setup 1* with an imbalance ratio $\rho = 100$ and ResNet18 architecture. The exploration covers a range from 10 to 50, systematically assessing the performance across various values of $C$ uses for IDW. Detailed results for different $C$ values are presented in Figure 12. We choose $C = 30$ for all five datasets based on its overall performance across the datasets.

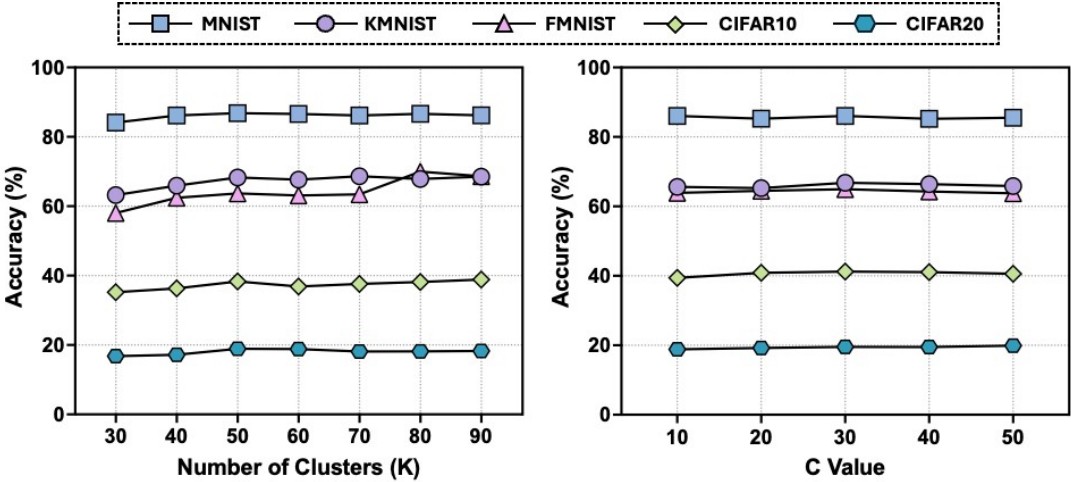

Figure 12: Ablation study analyzing the relationship between model performance and the number of clusters $K$ (**left**) and the $C$ value (**right**) across multiple datasets, using the S-NL+ICM method under *setup 1* with imbalance ratio $\rho = 100$ and ResNet18 architecture.

### D.5   Statistical Validation

In this subsection, we evaluate the statistical significance of the performance improvements achieved by our proposed method (ICM) over the original Mixup. We conduct experiments on CIFAR10, MNIST (*synthetic complementary labels*) and CLCIFAR10 (*real-world complementary labels*), running each configuration five times across several CLL algorithms, including S-NL, FWD, DM, and S-EXP. All experiments use the same hyperparameters described in the "Implementation Details" section, under both balanced and imbalanced settings. The resulting $p$-values are consistently well below 0.05 across all setups, providing strong evidence that ICM significantly outperforms the original Mixup. The corresponding results are shown in Figure 13, Figure 14, and Figure 15.

### D.6   Comparison of Model Architectures

We further extend our ablation study to evaluate the effectiveness of ICM under an imbalanced setting. Specifically, we train linear models, MLPs, and ResNet18 on MNIST with an imbalance ratio of $\rho = 100$. We observe the same trend as in the balanced setting: ICM consistently improves performance across all three architectures. The detailed results are illustrated in Figure 11.

Table 5: Effect of different encoders on ICM under the S-NL+ICM setting on CIFAR10.

| Encoders | Method | Dataset | Noise rate | Accuracy |
|----------|--------|---------|------------|----------|
| SimSiam | S-NL+ICM | CIFAR10 | **3.15%** | **79.13%** |
| MoCov3 | S-NL+ICM | CIFAR10 | 15.43% | 60.45% |
| BYOL | S-NL+ICM | CIFAR10 | 15.96% | 61.47% |

### D.7 Effect of Encoder Choice on ICM Performance

In the CLL setting, ordinary labels are unavailable or costly to obtain. As a result, the clustering induced by the encoder plays a critical role in ICM: it helps reduce label noise during data augmentation. By grouping samples into clusters, we encourage mixing between instances that are more likely to share the same true label. This increases the likelihood that the complementary-label condition $\bar{y}_i \in [K] \setminus y_i$ holds within each cluster, thereby reducing the risk of introducing additional noise. Intuitively, the better the encoder captures the underlying class structure, the less complementary-label noise is injected by ICM.

To investigate this, we conducted an ablation study comparing SimSiam with other self-supervised encoders, including MoCov3 (Chen et al., 2021) and BYOL (Richemond et al., 2020). We trained MoCov3 and BYOL with a ResNet18 backbone on CIFAR10 for 800 epochs, using the same hyperparameters as for SimSiam, to obtain pretrained encoders. At the representation-learning stage (*without labels*), the pretrained models achieve reasonably good performance for all three methods. However, when we plug these encoders into ICM, SimSiam yields substantially better performance than the other two. The results under the S-NL+ICM setting on CIFAR10 are summarized in Table 5.

We further analyzed this phenomenon by explicitly computing the noise rate introduced by ICM under different encoders. The results show that MoCov3 and BYOL induce substantially higher noise rates than SimSiam, which explains their lower accuracy when used within our framework. These findings indicate that ICM is indeed sensitive to the encoder, and that in our current implementation SimSiam provides the most suitable representations for reducing complementary-label noise and achieving strong performance.

## E Additional Results of the Experiments

In this section, we conduct additional experiments to cover the remaining datasets CIFAR10, CIFAR20 of *setup 1* and all remaining imbalance types from *setup 2* to *setup 3*. Tables 6 and 7 present the results for *setup 2* with imbalance ratios (*biased transition matrix*) $\rho = (3, 5, 10)$, while Tables 8 and 9 report the results for *setup 3*, which combines a biased transition matrix $\rho = (5, 10)$ with an ordinary imbalance distribution $\rho = (10, 50, 100)$. These findings consistently affirm that our proposed method outperforms other approaches, demonstrating superiority with SCL-NL, SCL-EXP, FWD-INT and DM loss functions across various imbalance settings.

Based on these experimental results, we propose utilizing ICM for simpler vision datasets such as MNIST, KMNIST, and FMNIST. For more complex vision datasets, including CIFAR10, CLCIFAR10, CIFAR20, and CLCIFAR20, MICM method is preferable to enhance the performance of the trained classifier.

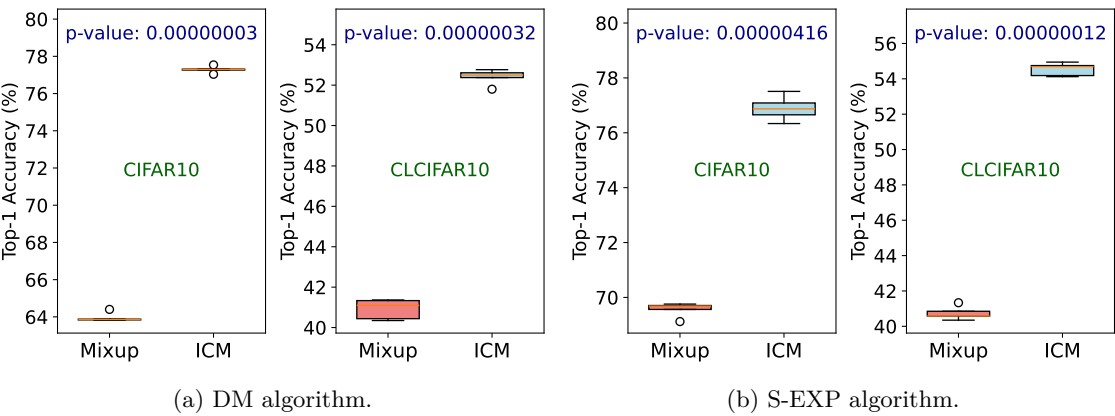

(a) DM algorithm.                                     (b) S-EXP algorithm.

Figure 13: Comparing the p-value of different between Mixup and ICM method on CIFAR10 and CLCIFAR10 with DM (**right**) and S-EXP (**left**) algorithms on both balanced and imbalanced ($\rho = 100$) scenarios.

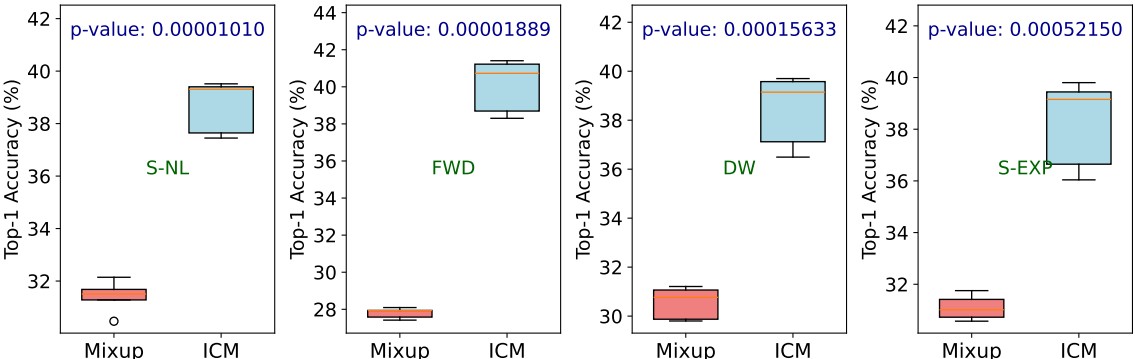

Figure 14: Comparing the p-value of different between Mixup and ICM method on CIFAR10 dataset under imbalanced ratio $\rho = 100$ with different algorithms.

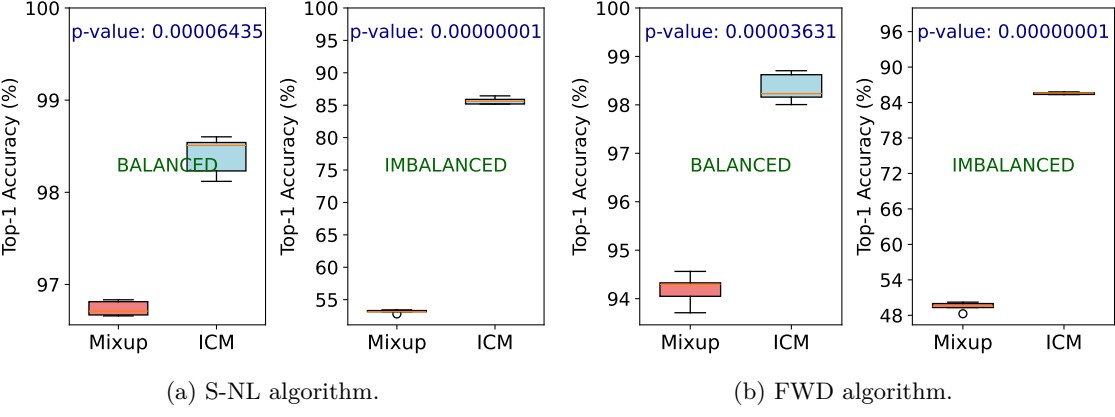

(a) S-NL algorithm.                                    (b) FWD algorithm.

Figure 15: Comparing the p-value of different between Mixup and ICM method on MNIST dataset with S-NL (**right**) and FWD (**left**) algorithms on both balanced and imbalanced ($\rho = 100$) scenarios.

Table 6: Top-1 validation accuracy (%) on *setup 2* with biased transition ratio $\rho = \{3, 5, 10\}$, $K = 50$, using S-NL, FWD, DM, and S-EXP losses on ResNet18 architecture

| Dataset | CIFAR10 | | | CIFAR20 | | |
|---|---|---|---|---|---|---|
| Method $\mid \rho$ | $\rho = 10$ | $\rho = 5$ | $\rho = 3$ | $\rho = 10$ | $\rho = 5$ | $\rho = 3$ |
| S-NL | $15.49_{0.24}$ | $19.08_{0.27}$ | $22.10_{0.30}$ | $6.70_{0.20}$ | $7.09_{0.39}$ | $8.16_{0.26}$ |
| S-NL+Mix | $20.74_{0.44}$ | $28.03_{0.32}$ | $33.71_{0.38}$ | $7.21_{0.36}$ | $9.83_{0.27}$ | $10.49_{0.45}$ |
| S-NL+ICM | $24.68_{0.42}$ | $33.97_{0.47}$ | $40.57_{0.45}$ | $7.36_{0.31}$ | $10.25_{0.25}$ | $10.65_{0.34}$ |
| S-NL+MICM | $\mathbf{26.16}_{0.26}$ | $\mathbf{34.96}_{0.31}$ | $\mathbf{42.66}_{0.36}$ | $\mathbf{8.73}_{0.27}$ | $\mathbf{11.97}_{0.39}$ | $\mathbf{14.81}_{0.23}$ |
| FWD | $57.57_{0.56}$ | $58.82_{0.47}$ | $60.80_{0.40}$ | $21.81_{0.31}$ | $21.63_{0.28}$ | $22.12_{0.29}$ |
| FWD+Mix | $64.10_{0.39}$ | $65.05_{0.49}$ | $64.94_{0.56}$ | $19.38_{0.37}$ | $22.03_{0.39}$ | $21.21_{0.28}$ |
| FWD+ICM | $78.97_{0.68}$ | $78.74_{0.47}$ | $78.70_{0.65}$ | $35.53_{0.43}$ | $37.34_{0.51}$ | $39.43_{0.55}$ |
| FWD+MICM | $\mathbf{80.00}_{0.55}$ | $\mathbf{80.61}_{0.36}$ | $\mathbf{80.70}_{0.63}$ | $\mathbf{42.82}_{0.61}$ | $\mathbf{45.70}_{0.37}$ | $\mathbf{45.67}_{0.52}$ |
| DM | $15.30_{0.28}$ | $16.04_{0.33}$ | $18.69_{0.32}$ | $6.13_{0.17}$ | $6.32_{0.34}$ | $6.81_{0.33}$ |
| DM+Mix | $26.15_{0.32}$ | $29.89_{0.28}$ | $33.28_{0.52}$ | $6.60_{0.19}$ | $8.17_{1.36}$ | $9.78_{0.35}$ |
| DM+ICM | $29.78_{0.08}$ | $36.94_{0.21}$ | $39.24_{0.14}$ | $7.06_{0.12}$ | $8.53_{0.21}$ | $10.16_{0.25}$ |
| DM+MICM | $\mathbf{31.06}_{0.09}$ | $\mathbf{39.47}_{0.12}$ | $\mathbf{41.40}_{0.21}$ | $\mathbf{8.04}_{0.16}$ | $\mathbf{10.68}_{0.24}$ | $\mathbf{13.84}_{0.21}$ |
| S-EXP | $15.52_{0.31}$ | $18.51_{0.29}$ | $21.79_{0.25}$ | $6.44_{0.21}$ | $7.00_{0.23}$ | $7.39_{0.24}$ |
| S-EXP+Mix | $20.99_{0.23}$ | $27.40_{0.27}$ | $31.06_{0.56}$ | $7.94_{0.43}$ | $9.37_{0.67}$ | $11.18_{0.37}$ |
| S-EXP+ICM | $24.30_{0.06}$ | $33.45_{0.20}$ | $40.36_{0.22}$ | $7.03_{0.27}$ | $10.14_{0.14}$ | $10.56_{0.03}$ |
| S-EXP+MICM | $\mathbf{25.34}_{0.08}$ | $\mathbf{34.35}_{0.14}$ | $\mathbf{41.56}_{0.19}$ | $\mathbf{8.27}_{0.10}$ | $\mathbf{10.68}_{0.12}$ | $\mathbf{14.44}_{0.09}$ |

Table 7: Top-1 validation accuracy (%) on *setup 2* with biased transition imbalance ratio $\rho = (3, 5, 10)$, $K = 50$, using S-NL, FWD, DM, and S-EXP losses with ResNet18 architecture

| Dataset | MNIST | | KMNIST | | FMNIST | |
|---|---|---|---|---|---|---|
| Method $\mid \rho$ | $\rho = 10$ | $\rho = 5$ | $\rho = 10$ | $\rho = 5$ | $\rho = 10$ | $\rho = 5$ |
| S-NL | $57.19_{0.38}$ | $66.47_{0.46}$ | $28.70_{0.44}$ | $41.41_{0.36}$ | $46.88_{0.54}$ | $67.73_{0.32}$ |
| S-NL+Mix | $35.77_{0.37}$ | $53.30_{0.35}$ | $23.42_{0.47}$ | $31.40_{0.45}$ | $34.95_{0.41}$ | $48.37_{0.36}$ |
| S-NL+ICM | $77.27_{0.72}$ | $\mathbf{97.99}_{0.68}$ | $54.84_{0.42}$ | $\mathbf{65.37}_{0.38}$ | $50.00_{0.39}$ | $64.06_{0.62}$ |
| S-NL+MICM | $\mathbf{95.67}_{0.56}$ | $96.99_{0.63}$ | $\mathbf{63.19}_{0.39}$ | $64.11_{0.46}$ | $\mathbf{54.98}_{0.57}$ | $\mathbf{67.28}_{0.62}$ |
| FWD | $93.00_{0.41}$ | $94.46_{0.59}$ | $65.67_{0.56}$ | $73.97_{0.37}$ | $80.60_{0.42}$ | $83.51_{0.58}$ |
| FWD+Mix | $88.70_{0.37}$ | $92.29_{0.46}$ | $63.88_{0.43}$ | $68.30_{0.39}$ | $78.37_{0.38}$ | $80.13_{0.31}$ |
| FWD+ICM | $\mathbf{98.15}_{0.45}$ | $\mathbf{98.21}_{0.35}$ | $\mathbf{88.09}_{0.68}$ | $\mathbf{87.75}_{0.47}$ | $\mathbf{85.95}_{0.54}$ | $\mathbf{85.80}_{0.48}$ |
| FWD+MICM | $98.00_{0.42}$ | $98.08_{0.78}$ | $86.87_{0.37}$ | $87.50_{0.41}$ | $85.02_{0.23}$ | $85.48_{0.57}$ |
| DM | $46.95_{0.32}$ | $70.38_{0.30}$ | $27.84_{0.41}$ | $32.71_{0.31}$ | $43.57_{0.39}$ | $54.03_{0.42}$ |
| DM+Mix | $35.39_{0.34}$ | $50.74_{0.36}$ | $26.06_{0.52}$ | $29.27_{0.58}$ | $39.56_{0.54}$ | $45.90_{0.49}$ |
| DM+ICM | $86.96_{0.12}$ | $\mathbf{97.01}_{0.11}$ | $\mathbf{63.65}_{0.23}$ | $\mathbf{65.46}_{0.21}$ | $52.42_{0.18}$ | $\mathbf{70.61}_{0.08}$ |
| DM+MICM | $\mathbf{92.99}_{0.14}$ | $94.85_{0.15}$ | $58.24_{0.21}$ | $60.43_{0.17}$ | $\mathbf{59.45}_{0.18}$ | $68.25_{0.12}$ |
| S-EXP | $57.21_{0.25}$ | $65.84_{0.28}$ | $28.71_{0.41}$ | $41.88_{0.43}$ | $46.08_{0.52}$ | $56.65_{0.34}$ |
| S-EXP+Mix | $27.91_{0.43}$ | $36.71_{0.45}$ | $22.16_{0.49}$ | $32.02_{0.72}$ | $43.95_{0.68}$ | $40.44_{0.56}$ |
| S-EXP+ICM | $69.11_{0.11}$ | $97.70_{0.19}$ | $54.26_{0.16}$ | $\mathbf{66.80}_{0.19}$ | $48.21_{0.14}$ | $63.32_{0.21}$ |
| S-EXP+MICM | $\mathbf{87.12}_{0.18}$ | $\mathbf{97.92}_{0.13}$ | $\mathbf{62.49}_{0.23}$ | $64.55_{0.16}$ | $\mathbf{56.38}_{0.20}$ | $\mathbf{66.47}_{0.13}$ |

Table 8: Top-1 validation accuracy (%) on *setup 3* combining biased transition ratio $\rho_1 = 5$ and long-tailed imbalance $\rho_2 = (10, 50, 100)$, $K = 50$, using S-NL, FWD, DM, S-EXP losses and ResNet18.

| Dataset | MNIST | | | CIFAR10 | | |
|---|---|---|---|---|---|---|
| Method $\mid \rho_1, \rho_2$ | 5, 100 | 5, 50 | 5, 10 | 5, 100 | 5, 50 | 5, 10 |
| S-NL | $39.18_{0.33}$ | $41.49_{0.37}$ | $51.76_{0.58}$ | $16.16_{0.39}$ | $14.84_{0.37}$ | $12.96_{0.36}$ |
| S-NL+Mix | $37.38_{0.35}$ | $37.89_{0.37}$ | $47.90_{0.39}$ | $16.00_{0.45}$ | $15.39_{0.38}$ | $17.15_{0.46}$ |
| S-NL+ICM | $89.44_{0.43}$ | $\mathbf{94.55}_{0.42}$ | $\mathbf{97.41}_{0.42}$ | $18.89_{0.49}$ | $21.00_{0.38}$ | $19.81_{0.45}$ |
| S-NL+MICM | $\mathbf{90.08}_{0.46}$ | $93.72_{0.51}$ | $96.80_{0.36}$ | $\mathbf{20.98}_{0.51}$ | $\mathbf{22.87}_{0.42}$ | $\mathbf{32.78}_{0.54}$ |
| FWD | $47.93_{0.34}$ | $54.02_{0.42}$ | $64.03_{0.30}$ | $22.18_{0.56}$ | $23.67_{0.47}$ | $23.94_{0.31}$ |
| FWD+Mix | $46.98_{0.31}$ | $51.23_{0.43}$ | $67.42_{0.48}$ | $30.20_{0.48}$ | $29.77_{0.43}$ | $38.68_{0.45}$ |
| FWD+ICM | $\mathbf{85.34}_{0.48}$ | $\mathbf{93.91}_{0.41}$ | $\mathbf{97.15}_{0.53}$ | $37.74_{0.36}$ | $40.34_{0.51}$ | $64.06_{0.42}$ |
| FWD+MICM | $83.49_{0.41}$ | $91.69_{0.49}$ | $95.79_{0.57}$ | $\mathbf{40.78}_{0.43}$ | $\mathbf{43.71}_{0.39}$ | $\mathbf{65.33}_{0.37}$ |
| DM | $36.91_{0.29}$ | $36.25_{0.51}$ | $48.57_{0.39}$ | $14.04_{0.41}$ | $14.62_{0.35}$ | $14.40_{0.28}$ |
| DM+Mix | $34.85_{0.32}$ | $40.68_{0.29}$ | $46.62_{0.43}$ | $18.26_{0.38}$ | $19.36_{0.54}$ | $18.37_{0.72}$ |
| DM+ICM | $\mathbf{88.13}_{0.12}$ | $\mathbf{91.38}_{0.21}$ | $\mathbf{96.75}_{0.15}$ | $21.17_{0.13}$ | $22.43_{0.07}$ | $32.55_{0.09}$ |
| DM+MICM | $81.66_{0.23}$ | $84.98_{0.15}$ | $92.73_{0.11}$ | $\mathbf{22.48}_{0.17}$ | $\mathbf{23.95}_{0.16}$ | $\mathbf{32.61}_{0.10}$ |
| S-EXP | $40.57_{0.32}$ | $40.46_{0.35}$ | $44.79_{0.30}$ | $15.68_{0.38}$ | $14.45_{0.27}$ | $12.80_{0.36}$ |
| S-EXP+Mix | $34.52_{0.41}$ | $36.08_{0.43}$ | $42.28_{0.52}$ | $15.58_{0.45}$ | $14.13_{0.57}$ | $16.58_{0.51}$ |
| S-EXP+ICM | $80.52_{0.09}$ | $84.18_{0.13}$ | $\mathbf{98.13}_{0.17}$ | $18.25_{0.20}$ | $19.78_{0.17}$ | $20.30_{0.16}$ |
| S-EXP+MICM | $\mathbf{83.17}_{0.16}$ | $\mathbf{85.11}_{0.21}$ | $97.82_{0.08}$ | $\mathbf{19.26}_{0.19}$ | $\mathbf{20.43}_{0.11}$ | $\mathbf{31.32}_{0.23}$ |

Table 9: Top-1 validation accuracy (%) on *setup 3* combining biased transition ratio $\rho_1 = 10$ and long-tailed imbalance $\rho_2 = (10, 50, 100)$, with $K = 50$, using S-NL, FWD, DM, S-EXP losses and ResNet18 architecture

| Dataset | MNIST | | | CIFAR10 | | |
|---|---|---|---|---|---|---|
| Method $\mid \rho_1, \rho_2$ | 10, 100 | 10, 50 | 10, 10 | 10, 100 | 10, 50 | 10, 10 |
| S-NL | $38.01_{0.40}$ | $41.19_{0.38}$ | $43.52_{0.42}$ | $11.60_{0.41}$ | $11.58_{0.53}$ | $11.82_{0.32}$ |
| S-NL+Mix | $33.13_{0.36}$ | $31.36_{0.36}$ | $32.22_{0.51}$ | $12.75_{0.47}$ | $13.88_{0.29}$ | $15.68_{0.57}$ |
| S-NL+ICM | $\mathbf{79.74}_{0.73}$ | $84.05_{0.46}$ | $\mathbf{97.15}_{0.53}$ | $16.11_{0.37}$ | $13.39_{0.49}$ | $14.55_{0.52}$ |
| S-NL+MICM | $71.07_{0.37}$ | $\mathbf{84.15}_{0.36}$ | $96.63_{0.43}$ | $\mathbf{16.85}_{0.48}$ | $\mathbf{18.90}_{0.42}$ | $\mathbf{16.93}_{0.51}$ |
| FWD | $38.48_{0.34}$ | $40.01_{0.53}$ | $62.91_{0.38}$ | $23.36_{0.27}$ | $22.44_{0.54}$ | $23.77_{0.42}$ |
| FWD+Mix | $47.35_{0.38}$ | $47.98_{0.43}$ | $60.35_{0.56}$ | $29.63_{0.36}$ | $31.75_{0.45}$ | $36.85_{0.52}$ |
| FWD+ICM | $\mathbf{84.13}_{0.41}$ | $\mathbf{90.95}_{0.45}$ | $\mathbf{96.72}_{0.71}$ | $37.01_{0.56}$ | $43.16_{0.64}$ | $62.88_{0.33}$ |
| FWD+MICM | $80.74_{0.63}$ | $87.63_{0.46}$ | $95.45_{0.46}$ | $\mathbf{38.02}_{0.42}$ | $\mathbf{46.17}_{0.52}$ | $\mathbf{63.81}_{0.45}$ |
| DM | $31.01_{0.27}$ | $32.38_{0.46}$ | $42.98_{0.29}$ | $15.98_{0.45}$ | $13.73_{0.37}$ | $13.02_{0.32}$ |
| DM+Mix | $26.42_{0.32}$ | $33.11_{0.35}$ | $37.51_{0.43}$ | $14.32_{0.56}$ | $14.59_{0.39}$ | $13.94_{0.34}$ |
| DM+ICM | $\mathbf{78.63}_{0.13}$ | $\mathbf{86.05}_{0.11}$ | $\mathbf{95.82}_{0.17}$ | $17.62_{0.07}$ | $19.45_{0.18}$ | $14.19_{0.14}$ |
| DM+MICM | $73.32_{0.09}$ | $80.54_{0.13}$ | $90.89_{0.12}$ | $\mathbf{19.35}_{0.07}$ | $\mathbf{20.86}_{0.16}$ | $\mathbf{17.44}_{0.10}$ |
| S-EXP | $37.83_{0.43}$ | $40.84_{0.52}$ | $44.72_{0.36}$ | $11.61_{0.41}$ | $11.89_{0.48}$ | $14.55_{0.37}$ |
| S-EXP+Mix | $33.59_{0.43}$ | $33.12_{0.34}$ | $33.85_{0.38}$ | $12.17_{0.31}$ | $12.71_{0.41}$ | $15.82_{0.36}$ |
| S-EXP+ICM | $\mathbf{73.06}_{0.06}$ | $\mathbf{76.24}_{0.09}$ | $79.23_{0.10}$ | $15.77_{0.15}$ | $14.01_{0.19}$ | $17.38_{0.12}$ |
| S-EXP+MICM | $71.28_{0.08}$ | $76.18_{0.11}$ | $\mathbf{79.27}_{0.12}$ | $\mathbf{16.71}_{0.09}$ | $\mathbf{16.30}_{0.13}$ | $\mathbf{20.26}_{0.16}$ |

