# OpenReview forum: "Intra-Cluster Mixup: An Effective Data Augmentation Technique for Complementary-Label Learning"
_TMLR — Accepted by TMLR_

### Review · Reviewer_NCsq · 2025-10-19

**Summary Of Contributions:**

In real-world scenarios, collecting high-quality labeled datasets is often difficult or even impossible. This paper introduces Intra-Cluster Mixup (ICM), which applies the Mixup method within samples belonging to the same cluster in the context of Complementary-Label Learning (CLL). As a result, the Mixup-induced label noise is significantly reduced from 16.24% to 0.95%. The authors also propose a Surrogate Complementary Loss (SCL) to minimize the likelihood associated with complementary labels, further improving model performance. Experiments were conducted on two CIFAR datasets, three MNIST variants, and two CLCIFAR datasets. The results show that ICM consistently improves performance across all datasets under both balanced and imbalanced settings.

**Additional Comments:**

No comments.

**Audience:**

Yes

**Audience Explanation:**

The findings are likely to interest researchers in weakly supervised learning, noisy label learning, and data augmentation.
This paper explores the reasons why Complementary-Label Learning (CLL) often fails. Although the proposed ICM does not achieve state-of-the-art performance, the authors clearly demonstrate that it improves model performance. These findings provide meaningful insights for addressing related challenges in areas such as semi-supervised learning, noisy-label learning, and class-imbalance learning.

**Broader Impact Concerns:**

No concerns.

**Claims And Evidence:**

Yes

**Claims Explanation:**

Strenghts
* The paper shows that ICM reduces this noise to 0.95%, verifying the mechanism behind the claimed improvement.
* Results across five synthetic datasets (CIFAR10, CIFAR20, MNIST, KMNIST, FMNIST) and two human-labeled datasets (CLCIFAR10, CLCIFAR20) consistently show performance gains across multiple complementary loss functions (SCL-NL, FWD, DM, EXP).

Weaknesses
* The performance improvement is clear, but the statistical significance (such as p-values) is not explicitly reported.
* The human-labeled datasets (CLCIFAR10 and CLCIFAR20) show relatively low accuracies, which might raise questions about the practical relevance.

**Requested Changes:**

1. Add statistical validation for the reported performance improvements. Presenting only the mean and standard deviation values is insufficient to determine whether the improvements are statistically significant.
2. Some datasets show relatively low accuracies, especially the human-labeled ones. It would be helpful for the authors to discuss how ICM could be applied or scaled to more practical scenarios and real-world settings.
3. Some tables (e.g., Table 2) are difficult to read due to the small text size. It could be more readable by splitting them into two smaller tables or enlarging the font size.

---

> ### Author Response · Authors · 2025-12-03
>
> Dear Reviewer NCsq,
>
> Thank you so much for your suggestion and valuable comments on our paper. Please find our response below.
>
> **Question 1**: Add statistical validation for the reported performance improvements. Presenting only the mean and standard deviation values is insufficient to determine whether the improvements are statistically significant.
>
> **Answer 1**: We are grateful to the reviewer for raising this important point.
> To thoroughly address this suggestion, we conducted an additional analysis on the CIFAR10 and CLCIFAR10, MNIST datasets under balanced and imbalanced settings. For each configuration, we ran five independent trials across several CLL algorithms, including S-NL, FWD, DM, and S-EXP, using the same hyperparameters (seed, learning rate, weight decay, etc.). We then computed p-values to assess whether the observed improvements are statistically significant. The results show that the p-values are well below $0.05$, providing strong evidence that our proposed method significantly outperforms the original Mixup.
>
> The results are presented in the following tables.
>
> | Method    | Dataset    | Data Type | P-value (one-sided paired t-test) |
> |-----------|------------|-----------|-----------------------------------|
> | S-NL+ICM  | CIFAR10    | Balanced  | 0.000002295237223                 |
> | FWD+ICM   | CIFAR10    | Balanced  | 0.000001871233427                 |
> | DM+ICM    | CIFAR10    | Balanced  | 0.0000002577924743                |
> | S-EXP+ICM | CIFAR10    | Balanced  | 0.00000416469561                  |
> | S-NL+ICM  | CLCIFAR10  | Balanced  | 0.000001771502372                 |
> | FWD+ICM   | CLCIFAR10  | Balanced  | 0.000003544954534                 |
> | DM+ICM    | CLCIFAR10  | Balanced  | 0.0000030158397743                |
> | S-EXP+ICM | CLCIFAR10  | Balanced  | 0.000003544954534                 |
> | S-NL+ICM  | MNIST      | Balanced  | 0.00006434519691                  |
> | FWD+ICM   | MNIST      | Balanced  | 0.000003631472677                 |
> | DM+ICM    | MNIST      | Balanced  | 0.00006668936969                  |
> | S-EXP+ICM | MNIST      | Balanced  | 0.002638035471                    |
>
> | Method    | Dataset | Data Type          | P-value (one-sided paired t-test) |
> |-----------|---------|--------------------|-----------------------------------|
> | S-NL+ICM  | CIFAR10 | Imbalanced (0.01)  | 0.00001010494369                  |
> | FWD+ICM   | CIFAR10 | Imbalanced (0.01)  | 0.00001889218244                  |
> | DM+ICM    | CIFAR10 | Imbalanced (0.01)  | 0.0001563338899                   |
> | S-EXP+ICM | CIFAR10 | Imbalanced (0.01)  | 0.0005215048447                   |
> | S-NL+ICM  | MNIST   | Imbalanced (0.01)  | 0.000000150977802                 |
> | FWD+ICM   | MNIST   | Imbalanced (0.01)  | 0.0000001066766908                |
> | DM+ICM    | MNIST   | Imbalanced (0.01)  | 0.0000006958821137                |
> | S-EXP+ICM | MNIST   | Imbalanced (0.01)  | 0.00000386481863                  |
>
> We also revised the manuscript to update the p-value statistics of our proposed method comparison to the baseline to reflect this in the paper.
> We added this result to the main paper and the Appendix D5.

---

> > ### Author Response · Authors · 2025-12-03
> >
> > **Question 2**: Some datasets show relatively low accuracies, especially the human-labeled ones. It would be helpful for the authors to discuss how ICM could be applied or scaled to more practical scenarios and real-world settings.
> >
> > **Answer 2**: This is a great question. Following your suggestion, we added a new paragraph to Section 4.2 “Results and Analysis”. In this new paragraph, we discussed how ICM could be applied and scaled to more practical settings.
> > In recent years, the field has observed that CLL does not seem fully practical yet, especially when moving from synthetic to real-world datasets and adding more classes. We found that the current state-of-the-art algorithm struggles under these conditions, highlighting the need for further work to make CLL more practical [1], [2]. Our proposed method (ICM) presents a novel data augmentation approach to make CLL more realistic. Specifically, ICM is designed to mitigate the effects of complementary-label noise associated with synthetic complementary samples. Through rigorous empirical evaluation, ICM demonstrates the effectiveness of encouraging complementary label sharing of nearby examples, leading to consistent performance improvements across a wide spectrum of experimental setups.
> > Our empirical experiments reveal that ICM substantially enhances the performance of learning models across a variety of state-of-the-art algorithms. In particular, when we applied our ICM method on a real-world dataset, CLCIFAR10, the model performance increased by 10%.
> > We hope our work contributes to empowering practitioners to develop more accurate and reliable models in real-world scenarios characterized by CLL.
> >
> > Reference
> >
> > [1] H.-H. Wang, M. T. Ha, N.-X. Ye, W.-I. Lin, and H.-T. Lin. CLImage: Human-annotated datasets for complementary-label learning. Transactions on Machine Learning Research, 2025. ISSN 2835-8856.
> >
> > [2] N.-X. Ye, T.-H. Mai, H.-H. Wang, W.-I. Lin, and H.-T. Lin. libcll: an extendable python toolkit for complementary-label learning, 2024. URL https://arxiv.org/abs/2411.12276.
> >
> > **Question 3**: Some tables (e.g., Table 2) are difficult to read due to the small text size. It could be more readable by splitting them into two smaller tables or enlarging the font size.
> >
> > **Answer 3**: Thanks for your reasonable suggestion. We have enlarged the font size of the text and changed the format to make it more readable. We have updated it in our manuscript.
> >
> > Additionally, we have resubmitted our revised manuscript incorporating your suggestions. Thank you!

---

### Review · Reviewer_eSLP · 2025-10-28

**Summary Of Contributions:**

This paper addresses the problem of data augmentation in Complementary-Label Learning (CLL), a setting where models are trained using labels that specify what a sample is not. The authors first demonstrate that the standard Mixup technique may be detrimental in CLL, as it introduces "complementary-label noise" that violates the CLL assumptions. To address this, they propose a two-stage method called Intra-Cluster Mixup (ICM). First, an offline step uses a pretrained self-supervised encoder to generate feature embeddings, which are then grouped using k-means clustering. Second, an online step applies Mixup only to pairs of samples drawn from within the same cluster. This strategy is designed for intra-class mixing, thereby reducing label noise. The empirical results show that ICM consistently improves the performance of several state-of-the-art CLL loss functions on various datasets.

**Audience:**

Yes

**Audience Explanation:**

It's interesting to introduce data augmentation methods into Complementary-Label Learning.

**Claims And Evidence:**

Yes

**Claims Explanation:**

1. The paper provides an experiment shown in Figure 1 demonstrating motivation clearly: that standard Mixup performs poorly, but an "oracle" version, Mixup Noise Free, is beneficial. This supports the claim that the noise introduced by Mixup is the specific cause of the performance degradation.
2. Empirical results on multiple datasets with both imbalanced and balanced cases show consistent improvement brought by ICM.
3. Table 2 shows ICM can be incorporated into several loss functions.
4. Gradient analysis gives further evidence.

**Requested Changes:**

1. I find that an additional method, MICM, is presented in the supplement as a superior method for complex and imbalanced datasets. Please clarify its existence and relationship to ICM in the main paper.
2. Would the performance of ICM be affected by different encoders? Do you try other encoders rather than SimSiam?

---

> ### Author Response · Authors · 2025-12-03
>
> Dear Reviewer eSLP,
>
> Thank you very much for your reasonable comments and constructive feedback.
> Please find our response below:
>
> **Question 1**: I find that an additional method, MICM, is presented in the supplement as a superior method for complex and imbalanced datasets. Please clarify its existence and relationship to ICM in the main paper.
>
> **Answer 1**: Thank you for your valuable comment. We have mentioned that briefly at the end of section 4.2. Following your suggestion, we expanded that further at the end section 4.2. We highlighted the revisions in blue text. We have resubmitted the revised version.
> We appreciate your valuable feedback to provide greater clarity and enrich the manuscript.
>
> **Question 2**: Would the performance of ICM be affected by different encoders? Do you try other encoders rather than SimSiam?
>
> **Answer 2**: Thank you for this reasonable suggestion. Indeed, the choice of encoder does affect the performance of ICM.
> In the CLL setting, ordinary labels are unavailable or costly to obtain. As a result, the clustering induced by the encoder plays a critical role in ICM: it helps reduce label noise during data augmentation. By grouping samples into clusters, we encourage mixing between instances that are more likely to share the same true label. This increases the likelihood that the complementary-label condition $\bar{y}_i \in [K] \setminus {y_i}$ holds within each cluster, thereby reducing the risk of introducing additional noise. Intuitively, the better the encoder captures the underlying class structure, the less complementary-label noise is injected by ICM.
>
> To investigate this, we conducted an ablation study comparing SimSiam with other self-supervised encoders, including MoCov3 and BYOL. We trained MoCov3 and BYOL with a ResNet18 backbone on CIFAR10 for 800 epochs, using the same hyperparameters as for SimSiam, to obtain pretrained encoders. At the representation-learning stage (without labels), the pretrained models achieve reasonably good performance for all three methods. However, when we plug these encoders into ICM, SimSiam yields substantially better performance than the other two. The results under the S-NL+ICM setting on CIFAR10 are summarized below:
>
>  Encoders | Method   | Dataset | Noise rate | Accuracry |
> |----------|----------|---------|------------|-----------|
> | SimSiam  | S-NL+ICM | CIFAR10 | **3.15**%      | 79.13 %   |
> | Mocov3   | S-NL+ICM | CIFAR10 | 15.43%     | 60.45 %   |
> | BYOL     | S-NL+ICM | CIFAR10 | 15.96%     | 61.47 %   |
>
> We further analyzed this phenomenon by explicitly computing the noise rate introduced by ICM under different encoders. The results show that MoCov3 and BYOL induce substantially higher noise rates than SimSiam, which explains their lower accuracy when used within our framework.
> These findings indicate that ICM is indeed sensitive to the encoder, and that in our current implementation SimSiam provides the most suitable representations for reducing complementary-label noise and achieving strong performance.
>
> We have added the ablation study in the main paper and Appendix **D.7** to support our claim and resubmitted the manuscript.

---

### Review · Reviewer_X2Lh · 2025-11-20

**Summary Of Contributions:**

The authors present several ideas:
 - analyzing why regular mixup do not work so well on complementary label learning (mostly attributing this to label noise)
 - propose a method to adress shortcomings found, where mixup is only done within feature based clusters
 - show that this method provides improved performance on some tasks, compared to regular mixup (higher top1 accuracy on CLCIFAR 10 and 20)
- provide evidence that this is due to lower noise. through a gradient analysis

**Audience:**

Yes

**Audience Explanation:**

Several application fields have high costs for aquiring data, such as the medical field. Producing labels is associated with costs, which leads to the idea of complementary label learning. Unfortunately, exactly such areas are also plagued with high costs for aquiring the unlabelled data as well (cases that needs biological samples, rare disease occurring etc), and this leads to need for data augmentation techniques. The authors seem to have made progress on the problem of combining these ideas.

The ideas of bias-variance decomposition of gradient esstimation is also an interesting idea and to me a novel one. This is also methodologicaly relevant.

**Broader Impact Concerns:**

I have no concerns of this sort.

**Claims And Evidence:**

No

**Claims Explanation:**

The basic idea is simple enough; do clustering of the data in feature space, and when training do mixup within each cluster only. Figure 2 conveys this idea well. From basic ideas about label noise (as given in page 4, *"Specifically, it is critical to minimize instances where $\bar{y}_i = y_j \vee \bar{y}_j = y_i$, as such occurences increase the error term $\varepsilon$ in equation 4*) this all makes sense. Table 2 shows that this method also provides improved performance. This far, all is good.

There are a couple of shortcomings, however. Mostly related to the presentation rather than the scientific content. This makes the paper hard to follow in some areas.

1) **Related works**. There are research on clustering methods applied in normal classification learning, and this is not acknowledged in the paper. One quick example I find after 2 minutes of searching online is https://aclanthology.org/2022.coling-1.333.pdf which has ideas very close to yours. This makes it hard to position the findings within the exiting litterature, and understand if this method is particular to complementary label learning, or if any mixup application would benefit from it.
2) **Undefined quntities**. The 'Noise Ratio' of Figure 1a (called Noisy Ratio in Figure 3) is not clearly defined. The variables $\varepsilon_i$ in equation (4) are not defined. The Iverson-style bracket notation used in equations 1-4 should be properly defined, since you use normal brackets in equation 15-17. The parameter $\rho$ that describes class imbalance is not defined. The 'Mixup Noise Free' method referenced in figure 1 and other places is not defined clearly. Was that a previous name for the ICM method?
3) **Notation**. $f$ is a classifier in section 2.1, and $g$ is a score function. In equation 1-4 it seems $g$ is the classifier. It is called 'model' in Definition 2. On page 6 you tell us that $\lambda_x = \lambda_\bar{y}$ so why use to different symbols if they are the same anyways? In algorithm 1, line 12-14 has roman lowercase numbered steps. Why not use that for line 10, 11 and 15? In equations 7 and 8, $g$ has a subscript $\theta$ tha is never used elsewhere.
4) **Unclear findings**. In figure 4a, cutout seems to hurt the performance of ICM. This is interesting and should be commented on somewhere. it was surprising to me. In section 6 you mention how ICM works with linear models and MLPs, but you have not presented any findings on this in the main text.

**Requested Changes:**

Critical changes:

1. Ensure that all notation is consistent and that all quantities are defined.
2. Discuss embedding based selective mixup in general (not only complementary label learning()

Non-critical changes:

3. I think you should work through the whole paper to ensure it is coherent. Ensure notational consistency throughout. Make sure that things are named properly and consistently between body text and figures/tables. Combine section 3.2 and 3.3 into one section so they work together as a whole. Place Figure 2 earlier in the article so the reader has the main message with them while reading the rest.
4. I think it would be very interesting to add a showcase of ICM on a regular classification problem, not only complementary labels.

---

> ### Author Response · Authors · 2025-12-03
>
> Dear Reviewer X2Lh,
>
> Thank you so much for all your suggestions and constructive comments. Please find our response below.
>
> **Question 1**: *Related works*. There is research on clustering methods applied in normal classification learning, and this is not acknowledged in the paper. One quick example I find after 2 minutes of searching online is https://aclanthology.org/2022.coling-1.333.pdf, which has ideas very close to yours. This makes it hard to position the findings within the existing literature, and understand if this method is particular to complementary label learning or if any mixup application would benefit from it.
>
> **Answer 1**: Thank you for pointing this out.
> We have checked the example above. The key difference in the paper above and our work is not only a different setup but also a design difference. The above paper used Mixup in ordinary label learning in NLP, with the key idea being distance-based sample selection. Our work used Mixup in complementary-label learning (CLL), with the key idea of Mixup within clustering.
>
> We have done a more thorough literature survey, including all clustering with Mixux [1], [2], [3], but we have done something different among these works, from design methodology and different problem settings. To the best of our knowledge, no other works have been done on the idea of K-means and Intra-Cluster Mixup in CLL before.  We are also the first work to try applying data augmentation in CLL.
>
> Reference
>
> [1] Xie, X., Yangning, L., Chen, W., Ouyang, K., Xie, Z., & Zheng, H. T. (2023, June). Global mixup: Eliminating ambiguity with clustering. In Proceedings of the AAAI Conference on Artificial Intelligence (Vol. 37, No. 11, pp. 13798-13806).
>
> [2] Navarro, M., & Segarra, S. (2023, June). GraphMAD: Graph mixup for data augmentation using data-driven convex clustering. In ICASSP 2023-2023 IEEE International Conference on Acoustics, Speech and Signal Processing (ICASSP) (pp. 1-5). IEEE.
>
> [3] Li, Z. and Jia, Y. Conmix: Contrastive mixup at representation level for long-tailed deep clustering. In The Thirteenth International Conference on Learning Representations, 2025.
>
> **Question 2**: *Undefined quantities*. The 'Noise Ratio' of Figure 1a (called Noisy Ratio in Figure 3) is not clearly defined. The variables $\epsilon$ in equation (4) are not defined. The Iverson-style bracket notation used in equations 1-4 should be properly defined, since you use normal brackets in equation 15-17. The parameter $\rho$ that describes class imbalance is not defined. The 'Mixup Noise Free' method referenced in figure 1 and other places is not defined clearly. Was that a previous name for the ICM method?
>
> **Answer 2**: Thank you for your very reasonable comments.
> We have fixed the inconsistent notation and defined the missing variable in the paper. We have resolved all your comments and clarified them by highlighting in blue part in the revised version.
> Extra-Class Mixup Noise-Free, Intra-Class Mixup Noise-Free are the versions of proof of concept before we proposed our method. We have revised our manuscript to improve the definition and make these points clearer for readers.
>
> **Question 3**: Notation. $f$ is a classifier in section 2.1, and g is a score function. In equation 1-4 it seems g is the classifier. It is called 'model' in Definition 2. On page 6 you tell us that $\lambda_x = \lambda_\bar{y}$ so why use to different symbols if they are the same anyways? In algorithm 1, line 12-14 has roman lowercase numbered steps. Why not use that for line 10, 11 and 15? In equations 7 and 8, $g$ has a subscript $\theta$ that is never used elsewhere.
>
> **Answer 3**: Thank you so much for your valuable comments.
> Following your suggestion, we have resolved all comments and updated the manuscript. We highlighted the revisions in blue text. We have resubmitted the revised version.

---

> > ### Author Response · Authors · 2025-12-03
> >
> > **Question 4**: Unclear findings. In figure 4a, cutout seems to hurt the performance of ICM. This is interesting and should be commented on somewhere. It was surprising to me. In section 6 you mention how ICM works with linear models and MLPs, but you have not presented any findings on this in the main text.
> >
> > **Answer 4**: Thank you for your suggestion. We have updated the comment to briefly explain why the cutout hurts performance in the main paper. We also recognized that we had missed the ablation study to support this claim. We have added the ablation study in the main paper and Appendix **D.6** to support our claim. The result of the ablation study is presented in the following tables.
> >
> > *Balance scenario*
> > | Model    | Method   | MNIST | KMNIST | FMNIST |
> > |----------|----------|-------|--------|--------|
> > | Resnet18 | S-NL     | 97.78 | 88.92  | 85.15  |
> > |          | S-NL+ICM | 98.20 | 89.09  | 85.25  |
> > |          | FWD      | 97.49 | 80.41  | 84.16  |
> > |          | FWD+ICM  | 98.18 | 88.92  | 84.75  |
> > |    MLP   | S-NL     | 94.20 | 74.72  | 84.56  |
> > |          | S-NL+ICM | 96.73 | 80.25  | 83.05  |
> > |          | FWD      | 93.09 | 68.22  | 84.59  |
> > |          | FWD+ICM  | 96.69 | 80.30  | 83.06  |
> > |  Linear  | S-NL     | 91.04 | 61.01  | 82.76  |
> > |          | S-NL+ICM | 92.30 | 63.01  | 81.15  |
> > |          | FWD      | 91.06 | 60.89  | 82.66  |
> > |          | FWD+ICM  | 92.28 | 63.00  | 81.24  |
> >
> > *Imbalance scenario*
> > | Model    | Method   | MNIST |
> > |----------|----------|-------|
> > | Resnet18 | S-NL     | 50.15 |
> > |          | S-NL+ICM | 85.83 |
> > |          | FWD      | 50.26 |
> > |          | FWD+ICM  | 85.27 |
> > | MLP      | S-NL     | 52.78 |
> > |          | S-NL+ICM | 74.40 |
> > |          | FWD      | 54.15 |
> > |          | FWD+ICM  | 74.38 |
> > | Linear   | S-NL     | 60.85 |
> > |          | S-NL+ICM | 77.15 |
> > |          | FWD      | 68.26 |
> > |          | FWD+ICM  | 77.14 |
> >
> > **Question 5**: Ensure that all notation is consistent and that all quantities are defined.
> >
> > **Answer 5**: Thank you so much for your comment. We have rechecked and solved all relevant notation inconsistencies in the paper. We resubmitted the revised version.
> >
> > **Question 6**: Discuss embedding-based selective mixup in general (not only complementary label learning)
> >
> > **Answer 6**: This is an interesting question. In the CLL setting, the true labels are not available during training. Therefore, we rely on an embedding-based approach to form clusters in the representation space, with the expectation that samples within the same cluster are likely to share the same (unknown) true label. This makes embedding-based CLL particularly suitable for selective Mixup, because the embeddings provide a surrogate notion of class similarity when explicit labels are missing.
> > In contrast, in the ordinary (fully supervised) setting, the true labels are available during training, so it is generally more natural and simpler to use these labels directly to guide Mixup pairing instead of relying on embeddings.
> > Our ablation study (Figure 2a in the revised manuscript) further illustrates this difference. In the CLL setting, applying Mixup only between samples that are inferred to belong to the same class (accuracy 85.46%) is more beneficial than applying Mixup across different classes (74.29%). However, in the ordinary-label setting, Mixup across different classes is more beneficial than Intra-Class Mixup (ICM), in both balanced and imbalanced scenarios. The detailed results of this ablation are provided in our response to Question 8.

---

> > > ### Author Response · Authors · 2025-12-03
> > >
> > > **Question 7**: I think you should work through the whole paper to ensure it is coherent. Ensure notational consistency throughout. Make sure that things are named properly and consistently between body text and figures/tables. Combine sections 3.2 and 3.3 into one section so they work together as a whole. Place Figure 2 earlier in the article so the reader has the main message with them while reading the rest.
> > >
> > > **Answer 7**: We appreciate your valuable feedback to provide greater clarity and enrich the manuscript. We have rechecked and solved all your comments. We combined sections 3.2 and 3.3 into one section and placed Figure 2 earlier in the article. We resubmitted the revised version.
> > >
> > > **Question 8**: I think it would be very interesting to add a showcase of ICM on a regular classification problem, not only complementary labels.
> > >
> > > **Answer 8**: Thank you for the insightful suggestion. To address it, we conducted an ablation study comparing ICM with original Mixup on a conventional supervised classification task, using ResNet18 on CIFAR10 under both balanced and imbalanced class distributions. In both settings, Mixup consistently outperforms ICM. This behavior is intuitive: original Mixup interpolates examples from different classes and therefore encourages sharing of features and labels across class boundaries, enriching the training signal and providing a stronger regularization effect. By contrast, ICM restricts interpolation to samples within the same class, so in the regular classification setting, it does not introduce new label combinations and contributes relatively limited additional information beyond standard data augmentation. The result of the ablation study is presented in the following table.
> > >
> > > | Method | Model    | Imbalance type   | Accuracy |
> > > |--------|----------|------------------|----------|
> > > | ICM    | Resnet18 | Balance          | 94.09%   |
> > > | Mixup  | Resnet18 | Balance          | **95.49**%   |
> > > | ICM    | Resnet18 | Imbalance (0.01) | 73.71%   |
> > > | Mixup  | Resnet18 | Imbalance (0.01) | **75.09**%   |

---

> > > > ### Comment · Reviewer_X2Lh · 2026-01-02
> > > >
> > > > Thank you for comprehensive answers. It seems you have done a thorough work.

---

> > > > > ### Author Response · Authors · 2026-01-03
> > > > >
> > > > > Thank you for the professional review and the encouraging words.

---

### Decision · Action_Editor_gygS · 2026-01-13

**Recommendation:** Accept as is

**Audience:**

Yes

**Audience Explanation:**

The findings of this paper belong to weakly supervised learning and I feel that many TMLR's audience would be interested in this topic.

**Claims And Evidence:**

Yes

**Claims Explanation:**

The authors demonstrate through controlled experiments that standard Mixup introduces harmful label noise in CLL, while their proposed Intra-Cluster Mixup (ICM) method significantly reduces this noise and delivers significant performance gains across multiple datasets, model architectures, and CLL loss functions.

---

> ### Author Response · Authors · 2026-01-22
> **Camera-Ready Version**
>
> Dear Action Editor,
>
> Thank you for your careful handling of our manuscript and for the constructive and insightful comments provided by you and the anonymous reviewers. We sincerely appreciate the time and effort invested in the review process, as well as the valuable feedback that helped us improve the clarity and rigor of our work.
>
> We have just submitted the camera-ready version for publication on the OpenReview system.
>
> Thank you again for your professionalism and support throughout the review process. We greatly appreciate the opportunity to publish our work in the journal.
>
> Sincerely,